# General Covariant Action Modeling: Constructing Generalized Manifolds via Spatio-Temporal Decoupling

Huaihai Lyu [1]  Chaofan Chen [1]  Mingyu Cao [2]  Yuheng Ji [1]  Changsheng Xu [1]

## Abstract

Achieving robust generalization from limited data is a central challenge in embodied intelligence. Prevailing methods fail by regressing absolute coordinates, which violates the principle of *general covariance*. Fundamentally, this conflates the intrinsic task geometry with rigid execution patterns, binding policies to specific motion styles and fixed speeds. To resolve this, we propose the **Generalized Action Manifold (GAM)** framework that enforces general covariance through structural disentanglement. Specifically, GAM realizes the manifold by enforcing invariance across two orthogonal dimensions: (1) Temporal Invariance, utilizing an *Arc-Length Parameterizer* to orthogonalize the spatial path geometry from temporal dynamics, ensuring robustness to velocity variations; (2) Geometric Invariance, where a *Schema-Affine-Factorization* mechanism maps trajectories to canonical "world lines" in a pose-normalized coordinate frame. This distinguishes invariant geometric schemas from affine modulations, ensuring spatial generalizability. By integrating GAM within a structured Vision-Language-Action (VLA) architecture, we enable sparse demonstrations to densely populate a continuous, valid action manifold. Empirical results demonstrate that GAM enables superior transfer and robustness capabilities, outperforming geometry-agnostic baselines.

## 1. Introduction

The success of foundation models (Wiggins & Tejani, 2022; Brown et al., 2020) stems largely from the unified representational abstractions, such as tokens (Achiam et al.,

[1]MAIS, Institute of Automation, Chinese Academy of Sciences. [2]Beijing Academy of Artificial Intelligence. Correspondence to: Chaofan Chen <chencfbupt@gmail.com>.

*Proceedings of the 43rd International Conference on Machine Learning*, Seoul, South Korea. PMLR 306, 2026. Copyright 2026 by the author(s).

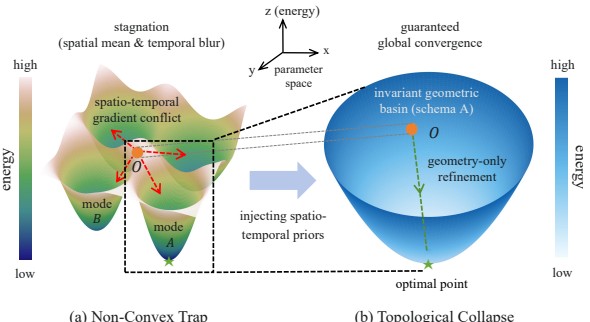

*Figure 1.* **The Optimization Landscape Transformation via GAM.** (a) The Non-Convex Trap: conditioned on the same observation, valid actions exhibit multi-modality in both geometry (*e.g.,* execution path) and dynamics (*e.g.,* execution speed). Direct regression averages these divergent signals, causing the optimization to stagnate at a high-energy saddle point. (b) Topological Collapse: our framework injects spatio-temporal priors to resolve this conflict. Temporal invariance decouples the execution speed, while geometric invariance locks the geometric intent.

2023) and patches (Dosovitskiy et al., 2020). While significant progress has been made in modeling the action modality (Zhao et al., 2023; Chi et al., 2023), establishing a theoretically grounded learning paradigm for VLA models remains an open challenge (Firoozi et al., 2025).

Current approaches typically treat robot actions as unstructured Euclidean vectors, training transformers to approximate conditional probability densities via direct regression (O'Neill et al., 2024; Octo Model Team et al., 2023; Kim et al., 2024). We argue that this paradigm is fundamentally flawed as it violates the principle of *General Covariance* (Ratliff et al., 2018), which implies that *task representations should be independent of the coordinate system*. Specifically, regressing absolute coordinates conflates the task's invariant geometry with instance-specific extrinsic frames and dynamics (Bronstein et al., 2021). As a result, a single geometric intent fractures into disjoint modes driven by task-extrinsic nuisance variations, including fluctuating execution speeds, shifting reference frames, and diverse spatial scales (Simeonov et al., 2022). This transforms learning into a non-convex optimization over a multi-modal landscape as visualized in Fig. 1(a), where valid behaviors cluster into sparse, locally convex *basins of attraction*(Calinon, 2020) separated by high-energy barriers.

Moreover, conflicting gradients from these divergent spatio-temporal basins cause the search to stagnate at high-energy saddle points (Dauphin et al., 2014), resulting in brittle, averaged policies that fail to generalize.

This optimization failure manifests as fundamental physical inconsistencies across both dimensions. Spatially, the phenomenon of *mode averaging* (Bishop, 1994; Florence et al., 2022) yields predictions that represent the mathematical mean of divergent paths, resulting in catastrophic failures such as grasping the empty space between two objects. Temporally, the reliance on absolute time indices forces the model to ignore natural variations in execution speed. This rigid temporal indexing introduces geometric misalignment between demonstrations of different speeds, which over-smooths critical dynamic features. Consequently, the learned policy lacks the geometric integrity to handle spatial and temporal perturbations.

To resolve these conflicts, we look beyond the Euclidean surface and investigate the intrinsic structure of the action manifold. Previous work (Calinon, 2020) has proposed that the effective action space is not a uniform hypersphere but is composed of sparse, locally convex basins of attraction. Crucially, while trajectories may appear distinct in absolute coordinates due to varying speeds or poses, valid demonstrations within the same basin are equivalent up to affine transformations of the spatial frame and reparametrizations of time. We refer to these invariant structures as ***action schemas***. Unlike rigid trajectories, a geometric schema captures the pure motion shape that is invariant to reference frames and scales. This conceptualization allows us to disentangle the invariant intent from instance-specific affine modulations and temporal profiles.

Building on this insight, we propose the **Generalized Action Manifold (GAM)**, a theoretical framework that enforces General Covariance through structural disentanglement. GAM constructs the generalized action manifold by enforcing invariance across two orthogonal dimensions. First, to ensure temporal invariance, we utilize an Arc-Length Parameterizer (ALP) to orthogonalize the spatial path geometry from its temporal dynamics. This effectively decouples the motion's "shape" from its "speed," eliminating temporal misalignment artifacts. Consequently, the learned policy captures high-frequency geometric details while remaining robust to velocity variations and system latencies. Complementarily, we achieve geometric invariance via the Schema-Affine-Factorization (SAF) mechanism. By mapping raw trajectories to canonical "World Lines" in a pose-normalized coordinate frame, SAF explicitly filters out spatial nuisance variables, such as absolute position and orientation. This decomposition distinguishes invariant *action schemas* from equivariant affine modulations. As visualized in Figure 1(b), GAM induces a topological col-

lapse: it first isolates the target solution basin (Mode $A$) via discrete schema selection, and then grounds the invariant schema into the physical world through parametric modulation. This effectively transforms the intractable global search into a well-posed local convex refinement problem. Furthermore, we instantiate GAM within a VLA architecture, transforming it from a black-box regressor into a structured generative agent. Empirical results demonstrate that this geometric rigor translates into superior generalization capabilities, outperforming geometry-agnostic baselines.

In summary, our contributions are threefold:

- We propose the **Generalized Action Manifold (GAM)** framework, a rigorous theoretical paradigm that enforces General Covariance across geometric invariance and temporal invariance.
- We show that the spatio-temporal priors reduce ambiguity by collapsing time-warped and pose-transformed demonstrations into locally convex schema basins, yielding a well-conditioned refinement objective.
- We demonstrate that GAM serves as a strong inductive bias for VLA models, achieving state-of-the-art performance and enabling generalization capabilities.

## 2. Related Work

### 2.1. Generative Modeling on Action Manifolds

Imitation learning models a multi-modal action distribution on a manifold $\mathcal{M}$. Behavior cloning regresses in Euclidean space and is therefore susceptible to mode averaging, which produces off-manifold actions (Pomerleau, 1988; Brohan et al., 2022; Bishop, 1994; Pathak et al., 2019). Energy-based and diffusion methods represent $\mathcal{M}$ implicitly but incur iterative inference (Florence et al., 2022; Chi et al., 2023; Pearce et al., 2023). Auto-regressive policies tokenize actions for efficient decoding (Reed et al., 2022; Brohan et al., 2023), yet common Euclidean quantizers (K-Means/binning) distort the local metric structure (Lee et al., 2024; Kim et al., 2024; Zhou et al., 2025). In contrast, GAM provides a structured manifold representation that preserves local metric consistency.

### 2.2. Geometric and Temporal Invariance in Robotics

Robust policies should respect geometric and temporal invariances. Prior work explores $SE(3)$-equivariance and Riemannian formulations (Bronstein et al., 2021; Simeonov et al., 2022; Ratliff et al., 2018), yet many generative policies still regress rotations directly and break manifold structure (Octo Model Team et al., 2023; Kim et al., 2024). In contrast, our approach maps actions to $\mathfrak{se}(3)$ and separates invariant geometry from equivariant parameters. For time, classical phase-based methods decouple shape and

speed (Ijspeert et al., 2013), while modern policies often bind trajectories to fixed timing (Zhao et al., 2023; Chi et al., 2023). GAM enforces a shape–phase separation via arc-length parameterization to improve robustness under temporal shifts.

## 3. Methodology

We first formalize the problem (Sec. 3.1) and analyze baseline failure modes (Sec. 3.2). Then we introduce the **Generalized Action Manifold (GAM)** framework (Sec. 3.3), and present structured VLA policy learning (Sec. 3.4).

### 3.1. Preliminaries

A VLA model learns a policy $\pi_\theta$ that generates action representations conditioned on visual observations $\mathbf{o}$ and language instructions $\mathbf{l}$. To capture temporal dependencies and ensure motion smoothness, the prediction target is typically an *action chunk* $\mathbf{a} \in \mathbb{R}^{T \times d}$, where $T$ denotes the action horizon and $d$ is the action dimensionality. The model is optimized to minimize the discrepancy between the predicted policy distribution and the ground-truth demonstrations, formulated as minimizing the expected negative log-likelihood:

$$\min_\theta \mathbb{E}_{(\mathbf{a},\mathbf{l},\mathbf{o}) \sim \mathcal{D}} \big[ -\log \pi_\theta(\mathbf{a} \mid \mathbf{o}, \mathbf{l}) \big]. \quad (1)$$

### 3.2. Theoretical Analysis: Manifold Factorization

We ground GAM in the geometry of the action manifold. Our central premise is that the observation–action mapping becomes multi-valued when *task-relevant geometric structure* is entangled with *nuisance symmetries* (e.g., spatial frames and temporal parameterizations). To formalize this, let $\mathcal{M}$ denote the set of feasible action trajectories. We define the symmetry group $\mathcal{G} = SE(3) \times \mathcal{T}$ acting on $\mathcal{M}$ by spatial reparameterization and temporal warping. The **shape space** is the quotient $\mathcal{Q} \triangleq \mathcal{M}/\mathcal{G}$ (*i.e.,* each $q \in \mathcal{Q}$ is an equivalence class of trajectories that share the same intrinsic action geometry but differ by a group element in $\mathcal{G}$). Accordingly, $\mathcal{M}$ can be viewed as a fiber bundle over $\mathcal{Q}$ with fiber $\mathcal{G}$ (locally, $\mathcal{M} \approx \mathcal{Q} \times \mathcal{G}$), where $\mathcal{Q}$ captures invariant task schemas and $\mathcal{G}$ encodes equivariant execution variables. In contrast, standard regression ignores this decomposition and learns in the ambient Euclidean coordinates, leading to unstable averaging across fibers.

#### 3.2.1. TEMPORAL MODE AVERAGING: THE NECESSITY OF DIFFEOMORPHISM INVARIANCE

**Problem Definition.** Temporal mode averaging arises because the standard Euclidean metric $L_2$ is sensitive to time parameterization. Consider a trajectory $\gamma(t) \in \mathcal{M}$ and its time-warped version $\gamma' = \gamma(\phi(t))$, where $\phi$ is a temporal diffeomorphism (representing speed variations). In the ambient space, $\|\gamma - \gamma'\|_2 > 0$. Consequently, when training on demonstrations with varying speeds, minimizing this error forces the model to learn the *arithmetic mean of "fast" and "slow" signals*, smoothing out high-frequency dynamics and critical contact transients.

**Theorem 1 (Diffeomorphism Invariance).** *Let $\mathcal{T}$ be the group of monotonic temporal diffeomorphisms. If a representation mapping $\Phi$ satisfies invariance under $\mathcal{T}$ (i.e., $\Phi(\gamma) = \Phi(\gamma \circ \phi)$ for all $\phi \in \mathcal{T}$), then the induced metric $d_\Phi(\gamma_1, \gamma_2) = \|\Phi(\gamma_1) - \Phi(\gamma_2)\|$ assigns zero distance to trajectories identical in geometry but distinct in speed. Under this metric, the regression target for a set of time-warped demonstrations collapses to a Dirac delta function, eliminating temporal variance.*

*Proof.* Consider a geometric path $\gamma(t)$ subject to random temporal jitter $\phi(t) = t + \delta(t)$, where $\delta(t)$ is zero-mean noise with variance $\sigma_t^2$. In the absence of invariance, the MSE objective minimizes the Euclidean distance between the model's prediction and the jittered observations. Consequently, this forces the estimator to converge to the arithmetic mean of trajectories executed at varying rates, thereby deviating from the true geometric shape (*e.g.,* cutting corners on high-curvature paths). By first-order Taylor expansion, the expected approximation error in the ambient space is bounded by:

$$\mathbb{E}\left[\|\gamma(t+\delta) - \gamma(t)\|^2\right] \approx \|\dot{\gamma}(t)\|^2 \cdot \sigma_t^2, \quad (2)$$

where $\dot{\gamma}(t)$ denotes the instantaneous velocity of the trajectory and $\sigma_t^2 = \mathbb{E}[\delta^2]$ represents the variance of the temporal misalignment. This reveals a critical pathology: the error is proportional to the squared velocity $\|\dot{\gamma}(t)\|^2$. In high-frequency regions (*e.g.,* impacts or sudden stops) where velocity changes rapidly, the error explodes. To minimize this objective, the model is forced to predict a trajectory with smaller derivatives ($\|\dot{\gamma}\| \to 0$), effectively acting as a *low-pass filter* that smooths out critical dynamics. Conversely, consider a representation mapping $\Phi$ that satisfies invariance under temporal diffeomorphisms, *i.e.,* $\Phi(\gamma(t)) = \Phi(\gamma(t+\delta))$. By definition, this mapping extracts intrinsic geometric properties independent of the execution time, implying that the sensitivity of the target to temporal jitter vanishes ($\frac{\partial \Phi}{\partial \delta} \equiv 0$). Consequently, the first-order Taylor expansion of the expected error collapses to zero: $\mathbb{E}_\delta \left\| \frac{\partial \Phi(\gamma(t))}{\partial \delta} \cdot \delta \right\|^2 = 0$. Since the variance of the target representation due to temporal jitter is zero, the probability density function of the target collapses to a single point mass, formally represented as a Dirac delta function $\delta_{\text{Dirac}}(\cdot - \Phi(\gamma))$. This proves that enforcing temporal invariance eliminates the irreducible error caused by misalignment. Unlike direct regression, the error bound becomes decoupled from the signal velocity $\|\dot{\gamma}(t)\|$, allowing the model to fit high-frequency geometric details without being

forced to apply low-pass smoothing. More details of proof can be found in Sec. A.1.1.

### 3.2.2. SPATIAL MODE AVERAGING: THE NECESSITY OF QUOTIENT CONVEXITY

**Problem Definition.** Mode averaging arises when the optimization landscape is non-convex (Bishop, 1994; Dauphin et al., 2014). In the ambient space $\mathbb{R}^D$, the valid action manifold is constructed as the union of group orbits under the total symmetry group $\mathcal{G}$. Formally, $\mathcal{M} = \bigcup_{g \in \mathcal{G}} g \cdot \mathcal{S}$, where each orbit represents a specific geometric intent instantiated under different spatial poses and temporal warps. Even for a single intent, this union is non-convex (*e.g.,* the Euclidean mean of a "left grasp" orbit and a "right grasp" orbit passes through the object, resulting in collision). To avoid this, the optimization must be performed in a space that satisfies *local convexity*.

**Theorem 2 (Convexity in Quotient Space).** *Let $\mathcal{M}$ be the action manifold invariant under the joint symmetry group $\mathcal{G}$. The optimization problem is multi-valued in the ambient space because the set of orbits is not closed under linear addition. However, by projecting the problem onto the Quotient Space $\mathcal{Q} = \mathcal{M}/\mathcal{G}$, the optimization landscape is simplified. We prove that for any two trajectories $\mathbf{a}_1, \mathbf{a}_2 \in \mathcal{M}$ belonging to the same geometric schema in $\mathcal{Q}$, their geodesic interpolation remains valid within $\mathcal{Q}$, whereas their linear interpolation in the ambient space exits $\mathcal{M}$.*

*Proof.* Consider two trajectories $\mathbf{a}_1, \mathbf{a}_2 \in \mathcal{M}$ representing the same intrinsic canonical shape $\mathbf{x}_c \in \mathcal{S}$ but lying on different orbits defined by group elements $g_1, g_2 \in \mathcal{G}$ (where $g = (\mathbf{T}, \phi)$ includes spatial transform and time warping). Assuming a left group action, we have $\mathbf{a}_1 = g_1 \cdot \mathbf{x}_c$ and $\mathbf{a}_2 = g_2 \cdot \mathbf{x}_c$. The standard regression objective inherently minimizes the Euclidean distance to the linear convex combination $\bar{\mathbf{a}} = \alpha \mathbf{a}_1 + (1 - \alpha)\mathbf{a}_2$. However, the symmetry group $\mathcal{G}$ is a non-linear manifold, and the linear interpolation of group elements $\bar{g} = \alpha g_1 + (1 - \alpha)g_2$ generally does not belong to $\mathcal{G}$. Taking the spatial rotation component $R \in SO(3)$ as a counterexample, the interpolated matrix $\bar{R} = \alpha R_1 + (1 - \alpha)R_2$ does not preserve orthogonality. Expanding the product:

$$\bar{R}^\top \bar{R} = (\alpha^2 + (1-\alpha)^2)I + \alpha(1-\alpha)(R_1^\top R_2 + R_2^\top R_1). \quad (3)$$

Since $R_1 \neq R_2$ are distinct rotations, for generic configurations $\bar{R}^\top \bar{R} \neq I$ and thus $\bar{R} \notin SO(3)$ except in degenerate cases. Consequently, there exists no valid group element $g \in \mathcal{G}$ such that $\bar{\mathbf{a}} = g \cdot \mathbf{x}_c$. Consequently, the mean $\bar{\mathbf{a}}$ falls into the "forbidden zone" off the manifold ($\bar{\mathbf{a}} \notin \mathcal{M}$), creating a saddle point in the optimization landscape. In contrast, GAM maps actions to the quotient space $\mathcal{Q}$ via the canonical projection $\mathcal{P} : \mathcal{M} \to \mathcal{Q}$. For both $\mathbf{a}_1$ and $\mathbf{a}_2$, this projection factors out the group orbit to yield the identical

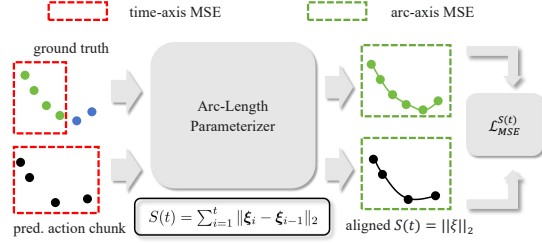

(a) Temporal Invariance: Arc-Length Parameterizer.

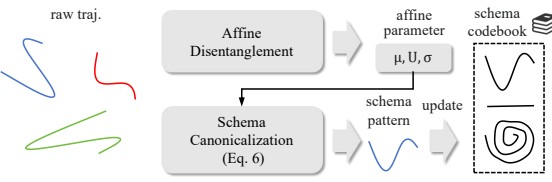

(b) Geometric Invariance: Schema-Affine Factorization.

*Figure 2.* **Disentangled Tokenization via GAM.** (a) Temporal Invariance: The Arc-Length Parameterizer transforms variable-speed trajectories into velocity-invariant geometric paths by re-indexing based on cumulative arc length. (b) Geometric Invariance: The Schema-Affine Factorization mechanism disentangles the spatial path, normalizing the trajectory into a canonical shape.

canonical shape $\mathcal{P}(\mathbf{a}_1) = \mathcal{P}(\mathbf{a}_2) = \mathbf{x}_c$. The optimization objective in $\mathcal{Q}$ collapses to minimizing the distance to a single target $\mathbf{x}_c$, defined as $\mathcal{L}_\mathcal{Q}(\hat{\mathbf{x}}) = \|\hat{\mathbf{x}} - \mathbf{x}_c\|^2$. Since $\nabla^2 \mathcal{L}_\mathcal{Q} = 2I \succ 0$, the landscape is strictly convex, ensuring a unique global optimum. More details of proof can be found in Sec. A.1.2.

### 3.3. Generalized Action Manifold

Guided by the theoretical analysis, our objective is to construct a generalized action representation that strictly satisfies two geometric properties: *temporal invariance* under diffeomorphisms to eliminate temporal blurring and *quotient convexity* under group symmetry to eliminate spatial mode averaging. To realize this, we propose the **Generalized Action Manifold (GAM)** framework, which constructs this rigorous representation by structurally disentangling the generation process via two orthogonal modules below.

### 3.3.1. ARC-LENGTH PARAMETERIZER

To satisfy Theorem 1, the learning objective must decouple the geometric path from the specific execution speed. We introduce a differentiable **Arc-Length Parameterizer (ALP)** mechanism directly within the training loop. Unlike standard MSE, which aligns trajectories via absolute timestamps (*i.e.,* coupling geometry and time), ALP aligns them via geometric progress. Specifically, given the discrete ground-truth waypoints $\{\boldsymbol{\xi}_{gt,k}\}_{k=0}^N$, we first compute the cumulative arc-length at each waypoint as $u_k = \sum_{j=1}^k \|\boldsymbol{\xi}_{gt,j} - \boldsymbol{\xi}_{gt,j-1}\|_2$ with $u_0 = 0$, which induces a piecewise-linear continuous spatial curve $\boldsymbol{\xi}_{gt}(u)$ parameterized by arc length

$u \in [0, u_N]$. During training, for a predicted action chunk $\hat{\boldsymbol{\xi}}$ of length $T$, we calculate its cumulative arc-length at each step $t$ analogously as $\hat{s}_t = \sum_{i=1}^{t} \|\hat{\boldsymbol{\xi}}_i - \hat{\boldsymbol{\xi}}_{i-1}\|_2$. We then dynamically truncate the ground truth to this predicted range and resample it via linear interpolation:

$$\tilde{\boldsymbol{\xi}}_{gt,t} = \text{LinearInterp}(\boldsymbol{\xi}_{gt}, \hat{s}_t), \quad \text{for } t = 1 \ldots T. \quad (4)$$

Here $\text{LinearInterp}(\boldsymbol{\xi}_{gt}, s)$ locates the segment $[u_k, u_{k+1}]$ containing the query $s$ and returns the convex combination $(1 - \alpha)\,\boldsymbol{\xi}_{gt,k} + \alpha\,\boldsymbol{\xi}_{gt,k+1}$ with $\alpha = (s - u_k)/(u_{k+1} - u_k)$, applied independently to translation, rotation, and gripper components. The procedure is fully differentiable with respect to $\hat{s}_t$, allowing gradients to flow back through the predicted arc length into the policy. This formulation ensures that the loss penalizes the deviation of the predicted point $\hat{\boldsymbol{\xi}}_t$ from the correct geometric path at the exact same distance traveled, regardless of the temporal execution speed.

### 3.3.2. SCHEMA-AFFINE FACTORIZATION

To satisfy Theorem 2, we must project the learning problem from the ambient manifold $\mathcal{M}$ onto the convex quotient space $\mathcal{Q} = \mathcal{M}/\mathcal{G}$. We propose the **Schema-Affine Factorization (SAF)** mechanism to explicitly perform this projection. SAF decomposes the spatial path into a spatially invariant geometric intent (action schema) and spatially equivariant execution parameters (affine transform).

For a trajectory $\gamma \in \mathcal{M}$, we denote its translational and rotational components by $\gamma = [\mathbf{p}, \mathbf{q}]$, where $\mathbf{p}$ represents the Cartesian translations and $\mathbf{q}$ represents the rotations. We utilize the spatial component $\mathbf{p}$ to analytically extract the optimal affine transformation parameters, effectively factoring out the $SE(3)$ group orbits. First, we compute the **translation** $\mu$ as the centroid of the trajectory positions: $\mu = \frac{1}{T} \sum_{t=1}^{T} \mathbf{p}_t$. Next, let $\mathbf{P} \in \mathbb{R}^{T \times 3}$ denote the matrix of trajectory positions whose $t$-th row is $\mathbf{p}_t^\top$. To determine a canonical orientation, we compute the covariance matrix of the centered positions and its eigendecomposition:

$$\mathbf{C} = (\mathbf{P} - \mathbf{1}\mu^\top)^\top (\mathbf{P} - \mathbf{1}\mu^\top) = U\Lambda U^\top, \quad (5)$$

where $\mathbf{1} \in \mathbb{R}^T$ is the all-ones vector. The columns of $U$ form a **canonical orthogonal frame** that aligns the trajectory's principal geometric axes with the reference coordinate system. Finally, the **scale** $\sigma$ is computed as the root-mean-square distance of positions from the centroid $\mu$.

By applying these parameters to the full trajectory (transforming both positions $\mathbf{p}$ and orientations $\mathbf{q}$), we obtain the *canonical shape* $\mathbf{x}_c$, defined as the concatenation of the normalized components:

$$\mathbf{x}_c = \begin{cases} \mathbf{p}_{can} = \sigma^{-1}(\mathbf{p} - \mu)U^{-1} \\ \mathbf{q}_{can} = U^{-1} \circ \mathbf{q} \end{cases} \quad (6)$$

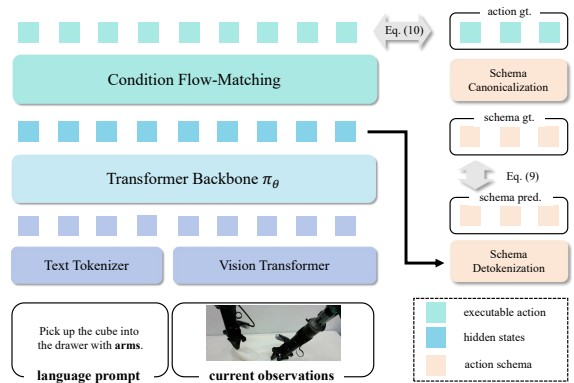

*Figure 3.* **Overview of GAM-VLA Architecture.** The GAM-VLA architecture integrates Vision and Language inputs into a structured prediction pipeline. (1) The hidden states predict the discrete action schema to lock the solution basin. (2) The Flow Head, conditioned on the schema, generates the fine-grained action signals. This hierarchical process guarantees the generation of valid, mode-consistent trajectories.

where $\circ$ denotes rotation composition. This ensures that the derived schema captures the intrinsic motion geometry invariant to the observer's reference frame.

This normalization process mathematically realizes the projection operator $\mathcal{P} : \mathcal{M} \to \mathcal{Q}$ described in Theorem 2. By factoring out $\mu, U$, and $\sigma$, we collapse the entire group orbit (*i.e.,* all possible spatial variations of the same action) into a single representative point $\mathbf{x}_c$ in the quotient space. In this normalized coordinate system, the non-linear constraints imposed by the group structure are factored out. Consequently, the local neighborhood of $\mathbf{x}_c$ becomes a linear vector space where the Euclidean metric is valid. This ensures that the optimization landscape for the subsequent schema matching is locally convex, effectively eliminating the saddle points caused by spatial misalignment.

### 3.4. GAM-VLA Policy Learning

We instantiate GAM within a VLA architecture using a two-stage strategy. We first construct the geometric prior from offline data analytically, and then train the policy to align visual-linguistic inputs with this structure via a *Plan-then-Execute* paradigm.

### 3.4.1. UNSUPERVISED MANIFOLD TOKENIZATION

Instead of learning a neural tokenizer from raw actions, we employ an analytical approach to first disentangle the intrinsic geometry. For each trajectory in the dataset, we apply SAF to extract the ground-truth geometric signals (*e.g.,* affine parameters $(\mu, U, \sigma)$ and the canonical shape $\mathbf{x}_c$). Subsequently, to establish the discrete action schemas, we treat the set of extracted canonical shapes as a dataset to train a Residual Vector Quantized Variational Autoencoder (RVQ-VAE). The RVQ-VAE consists of an encoder $E_{tok}$, a

*Table 1.* **Experimental Results for the LIBERO Benchmarks.**

| Model | Spatial | | Object | | Goal | | Long | | Average | |
|---|---|---|---|---|---|---|---|---|---|---|
| | SR ($\uparrow$) | Rank ($\downarrow$) | SR ($\uparrow$) | Rank ($\downarrow$) | SR ($\uparrow$) | Rank ($\downarrow$) | SR ($\uparrow$) | Rank ($\downarrow$) | SR ($\uparrow$) | Rank ($\downarrow$) |
| Octo (Octo Model Team et al., 2023) | 78.9% | 11 | 85.7% | 7 | 84.6% | 5 | 51.1% | 7 | 75.1% | 7 |
| OpenVLA (Kim et al., 2024) | 84.7% | 6 | 88.4% | 6 | 79.2% | 6 | 53.7% | 6 | 76.5% | 6 |
| SpatialVLA (Qu et al., 2025) | 88.2% | 5 | 89.9% | 5 | 78.6% | 7 | 55.5% | 5 | 78.1% | 5 |
| $\pi_0$ (Black et al., 2024) | 96.8% | 2 | **98.8%** | 1 | 95.8% | 2 | 85.2% | 3 | 94.1% | 2 |
| $\pi_0$-FAST (Pertsch et al., 2025) | 96.4% | 3 | 96.8% | 4 | 88.6% | 4 | 60.2% | 4 | 85.5% | 4 |
| BEAST (Zhou et al., 2025) | 92.9% | 4 | 97.5% | 2 | 93.1% | 3 | 86.4% | 2 | 92.5% | 3 |
| **GAM (ours)** | **98.6%** | 1 | 97.5% | 2 | **96.3%** | 1 | **92.0%** | 1 | **96.1%** | 1 |

*Table 2.* **Evaluation on SimplerEnv–WidowX across diverse manipulation tasks.**

| Model | Put Spoon | | Put Carrot | | Stack Block | | Put Eggplant | | Overall |
|---|---|---|---|---|---|---|---|---|---|
| | Grasp | Success | Grasp | Success | Grasp | Success | Grasp | Success | Success |
| Octo-Base (Octo Model Team et al., 2023) | 34.7% | 12.5% | 52.8% | 8.3% | 31.9% | 0.0% | 66.7% | 43.1% | 16.0% |
| $\pi_0$ (Black et al., 2024) | - | 29.1% | - | 0.0% | - | 16.6% | - | 62.5% | 27.1% |
| Octo-Small (Octo Model Team et al., 2023) | 77.8% | 47.2% | 27.8% | 9.7% | 40.3% | 4.2% | 87.5% | 56.9% | 29.5% |
| RoboVLMs (Li et al., 2024b) | 70.8% | 45.8% | 33.3% | 20.8% | 54.2% | 4.2% | 91.7% | 79.2% | 37.5% |
| OpenVLA-OFT (Kim et al., 2025) | - | 34.5% | - | 30.0% | - | 30.0% | - | 72.5% | 41.8% |
| SpatialVLA (Qu et al., 2025) | 20.8% | 16.7% | 29.2% | 25.0% | 62.5% | 29.2% | 100% | 100% | 42.7% |
| $\pi_0$-FAST (Pertsch et al., 2025) | 33.3% | 29.2% | 25.0% | 20.8% | 37.5% | 12.5% | 75.0% | 66.7% | 32.3% |
| BEAST (Zhou et al., 2025) | 66.7% | 41.7% | 37.5% | 25.0% | 50.0% | 20.8% | 87.5% | 75.0% | 37.5% |
| **GAM (ours)** | **91.7%** | **58.3%** | **66.7%** | **37.5%** | **79.2%** | **41.7%** | **95.8%** | **87.5%** | **56.3%** |

learnable codebook $\mathcal{C} = \{\mathbf{e}_k\}_{k=1}^K$, and a decoder $D_{tok}$. The encoder maps the canonical shape to a latent vector $\mathbf{z}_e = E_{tok}(\mathbf{x}_c)$, which is then quantized to the nearest codebook vector $\mathbf{e}_{z_s}$:

$$z_s = \arg\min_k \|\mathbf{z}_e - \mathbf{e}_k\|_2. \quad (7)$$

The model is trained to minimize the reconstruction error of the canonical shape and the commitment error of the latent representation with a weight coefficient $\beta = 1.0$:

$$\mathbb{E}_{\mathbf{x}_c \sim \mathcal{D}}[\|\mathbf{x}_c - D_{tok}(\mathbf{e}_{z_s})\|^2 + \beta\|\operatorname{sg}[\mathbf{z}_e] - \mathbf{e}_{z_s}\|^2], \quad (8)$$

where $\operatorname{sg}[\cdot]$ denotes the stop-gradient operator. Upon convergence, the learned codebook $\mathcal{C}$ captures the prototypical geometric primitives of the action space. Each trajectory is then automatically annotated with the discrete schema index $z_s^{gt}$, which serves as the geometric supervisor for the downstream policy.

### 3.4.2. STRUCTURED CONDITION FLOW

Building upon the extracted schemas, the architecture consists of a transformer backbone to predict the action schema and a conditional flow head to infer executable actions as shown in Fig. 3. The *Planner* acts as the decision-maker, resolving multi-modal ambiguity by locking the geometric intent and estimating affine constraints. The *Executor* functions as the trajectory generator conditioned on the *action intent hidden state* derived from the Planner. This design ensures that the Executor's continuous generation process is confined within the specific local convex basin selected by the Planner, thereby translating discrete semantic plans into continuous physical control.

**Planner: Intent Alignment.** The backbone functions as a high-level planner $\pi_\theta$. It takes visual-linguistic inputs $(\mathbf{o}, \mathbf{l})$

and outputs a categorical distribution over the schema space. We supervise this process using the **SAF-derived discrete intents** $z_s^{gt}$. The planner minimizes the cross-entropy loss to lock the correct solution basin:

$$\mathcal{L}_{plan} = -\mathbb{E}_{(\mathbf{o}, \mathbf{l}, z_s^{gt}) \sim \mathcal{D}} \left[ \log \pi_\theta(z_s^{gt} \mid \mathbf{o}, \mathbf{l}) \right]. \quad (9)$$

**Executor: Conditional Flow Matching.** To generate high-fidelity actions, we attach a lightweight flow matching head. Conditioned on the *action intent hidden state* $\mathbf{h}_{intent}$ extracted from the planner, the Executor generates the executable action chunk $\hat{\mathbf{a}}$ directly. We formulate the generation as a standard flow-matching paradigm, transporting Gaussian noise $p_0$ to the data distribution $p_1$. To enforce *temporal invariance* during training, we do not use the raw time-indexed ground truth as the target $x_1$. Instead, we construct a *prediction-conditioned, arc-length-aligned* target $\boldsymbol{\xi}_{gt}^{alp}$ by interpolating the ground-truth trajectory at the predicted cumulative arc-length positions according to Eq. 4. This ensures the flow head learns the pure geometric path, independent of the demonstration speed. We optimize the vector field $v_t$ using a composite objective:

$$\mathcal{L}_{exec} = \mathbb{E}_{t, \mathbf{x}_0, \boldsymbol{\xi}_{gt}}[\|v_t - (\boldsymbol{\xi}_{gt}^{alp} - \mathbf{x}_0)\|^2] + \lambda\mathcal{L}_{vel}, \quad (10)$$

where $\mathbf{x}_0 \sim \mathcal{N}(\mathbf{0}, \mathbf{I})$ denotes the Gaussian noise sampled as the source endpoint of the flow matching path, and $\lambda$ is the weight coefficient balancing flow matching against velocity regularization. The term $\mathcal{L}_{exec}$ aligns the generation with the geometry of the ground truth. However, since ALP removes time information, the model might converge to trivial stationary solutions (*i.e.,* infinite slowness). To prevent this, we introduce a *velocity regularization* term $\mathcal{L}_{vel}$ that encourages the generated trajectory to maintain a

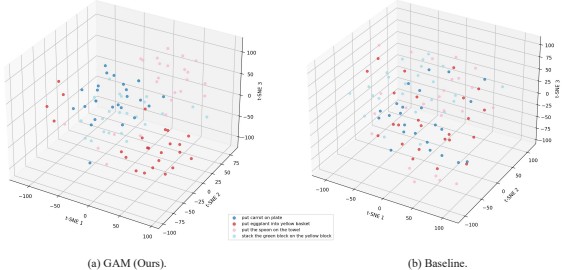

*Figure 4.* **t-SNE Visualization of Hidden States.**

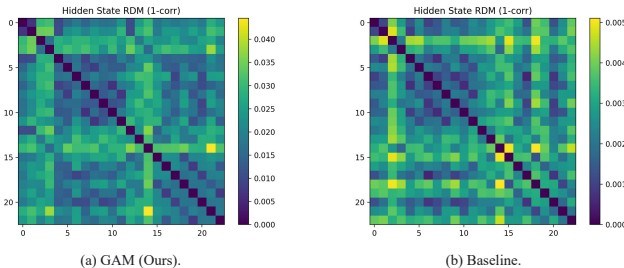

*Figure 5.* **Representational Similarity Analysis.**

spatial stride consistent with the demonstration:

$$\mathcal{L}_{\text{vel}} = \frac{1}{T-1} \sum_{i=1}^{T-1} \left( \|\hat{\boldsymbol{\xi}}_i - \hat{\boldsymbol{\xi}}_{i-1}\|_2 - \|\boldsymbol{\xi}_{gt,i} - \boldsymbol{\xi}_{gt,i-1}\|_2 \right)^2. \tag{11}$$

This penalizes deviations between the predicted and demonstrated step sizes, preventing the model from collapsing to trivial stationary solutions while preserving the natural execution rhythm.

## 4. Experiments

This section evaluates the effectiveness of the **Generalized Action Manifold (GAM)** framework. We evaluate whether GAM can construct a continuous, valid action manifold from sparse demonstrations. We structure our evaluation around three research questions (RQ):

*RQ1: Manifold Construction for Generalist Learning.* Does the globally consistent manifold constructed by GAM translate into superior performance on long-horizon, spatially diverse, and goal-conditioned tasks?

*RQ2: Validation of General Covariance.* Can GAM empirically resolve mode averaging and achieve zero-shot adaptation to spatial and temporal perturbations compared to geometry-agnostic baselines?

*RQ3: Ablation Study.* Are both geometric modules (SAF and ALP) and both training objectives ($\mathcal{L}_{\text{plan}}$, $\mathcal{L}_{\text{vel}}$) individually necessary for the observed performance gains?

### 4.1. Implementation Details

In real-world experiments, the VLA backbone $\pi_\theta$ is a purely autoregressive Transformer with **3B** parameters, initialized from **Qwen2.5-VL**. The raw action horizon is set to 30 frames. For the GAM tokenizer, we pre-train on the dataset (mixture with LIBERO, Bridge and RT-1) for 5 epochs to learn the geometric schemas and whitening statistics. For VLA fine-tuning, we employ a dual-head architecture (Discrete Schema Head + Continuous Modulation/Residual Head) and train with a batch size of 256 on 8×A800 GPUs for 10 epochs. The velocity regularization weight in Eq. 10 is set to $\lambda = 2.0$. Detailed hyperparameters for the SAF

clustering and ALP time-warping are provided in Sec. A.2.

### 4.2. Benchmarks

To evaluate the quality of the constructed manifold, we use the full **LIBERO** (Liu et al., 2024) suite. Beyond standard full-training evaluation, we treat LIBERO-Long (sequencing 10 sub-tasks) as a proxy for global manifold consistency, and examine how it relates to more specific capabilities on LIBERO-Spatial, Object, and Goal. **SimplerEnv** (Li et al., 2024a) assesses robustness under visual distribution shifts and camera viewpoint changes, probing the decoupling between visual perception and geometric execution. Comprehensive task definitions and environment settings are provided in Sec. A.4, and all evaluation metrics are summarized in Sec. A.6.

### 4.3. RQ1: Manifold Construction for Generalist Learning

Table 1 summarizes the success rates across the **LIBERO** task suites. We compare GAM against strong baselines including diffusion-based Octo (Octo Model Team et al., 2023), AR-based OpenVLA (Kim et al., 2024), and specialized tokenizers such as FAST (Pertsch et al., 2025) and BEAST (Zhou et al., 2025). GAM achieves the highest average success rate of **96.1%**, significantly outperforming the top baseline $\pi_0$ (94.1%) and BEAST (92.5%). This empirical superiority confirms that manifold-aligned tokenization provides a more robust foundation for generalist policies than geometry-agnostic approaches. Notably, on the challenging **LIBERO-Long** suite, GAM achieves a remarkable **92.0%** success rate. We attribute this stability to the *Topological Collapse* induced by our SAF mechanism: by locking the high-level intent into discrete schemas, the policy maintains long-horizon consistency without drifting, effectively transforming the complex planning problem into a sequence of stable local refinements. Table 2 reports performance under significant domain shifts in **SimplerEnv**, designed to test the limits of generalization. GAM achieves a state-of-the-art overall success rate of **56.3%**, surpassing the strongest baseline SpatialVLA (42.7%) by a large margin of **+13.6%**. The performance gap is particularly pronounced on tasks involving severe visual shifts, such as

*Table 3.* **Ablations of GAM Components.**

*(a)* **Effects of Manifold Alignment.**

| SAF | ALP | Spatial | Object | Goal | Long |
|---|---|---|---|---|---|
| ✗ | ✗ | 92.0 | 91.5 | 92.7 | 82.4 |
| ✗ | ✓ | 91.6 | 93.4 | **96.8** | 76.4 |
| ✓ | ✗ | 92.8 | 97.3 | 90.0 | 89.1 |
| ✓ | ✓ | **98.6** | **97.5** | 96.3 | **92.0** |

*(b)* **Effects of Training Objectives.**

| $\mathcal{L}_{plan}$ | $\mathcal{L}_{vel}$ | Spatial | Object | Goal | Long |
|---|---|---|---|---|---|
| ✓ | ✗ | 93.6 | 96.8 | 95.2 | 87.3 |
| ✗ | ✓ | 52.7 | 18.2 | 31.6 | 5.1 |
| ✓ | ✓ | **98.6** | **97.5** | **96.3** | **92.0** |

*Table 4.* **Validation of Temporal Invariance via ALP.**

| $\delta$ | ALP | Put Spoon | Put Carrot | Stack Block | Put Eggplant |
|---|---|---|---|---|---|
| 0.0 | ✗ | 45.8% | 33.3% | 12.5% | 41.7% |
| | ✓ | **58.3%** | **37.5%** | **41.7%** | **87.5%** |
| 0.5 | ✗ | 29.2% | 25.0% | 18.2% | 29.2% |
| | ✓ | **45.8%** | **41.7%** | **20.8%** | **70.8%** |
| 1.0 | ✗ | 12.5% | 16.7% | 4.2% | 33.3% |
| | ✓ | **33.3%** | **45.8%** | **25.0%** | **62.5%** |

Put Eggplant (95.8% Grasp success). This demonstrates that GAM effectively enforces Visual-Geometric Decoupling: the Schema Head captures the invariant task logic, while the Modulation Head adapts to the shifted visual frame via affine transformations.

### 4.4. RQ2: Validation of General Covariance

To empirically verify the theoretical claims of General Covariance, we conduct targeted stress tests to isolate the contributions of Temporal Invariance (via ALP) and Geometric Invariance (via SAF).

**Validation of Temporal Invariance.** We evaluate robustness to execution-speed variations on SimplerEnv by injecting stochastic speed noise during training. We sample a global scaling factor $\beta \sim U[-\delta, \delta]$ and use $\alpha = e^{\beta}$ as a multiplicative speed factor to rescale the time axis before resampling actions. As shown in Table 4, the baseline (w/o ALP) degrades sharply as jitter increases: for PutSpoon, success drops from 45.8% at $\delta = 0$ to 12.5% at $\delta = 1.0$; for PutCarrot, from 33.3% to 16.7%; and for StackBlock, 12.5% to 4.2%. In contrast, ALP remains substantially more stable under the same shift: at $\delta = 1.0$, it achieves 33.3% on Put Spoon (vs. 12.5%), 45.8% on Put Carrot (vs. 16.7%), and 25.0% on Stack Block (vs. 4.2%), with a similarly large margin on Put Eggplant (62.5% vs. 33.3%). These consistent gains under large speed perturbations indicate that ALP decouples geometric path learning from temporal scaling, preventing velocity-profile averaging and over-smoothing.

**Validation of Geometric Invariance.** We investigate the structure of the learned latent space to verify whether SAF successfully disentangles geometric intent from spatial execution. First, we visualize the t-SNE projection of hidden states from four distinct tasks from SimplerEnv in Fig. 4. The Baseline (b) exhibits entangled representations with significant overlap between tasks, indicating a failure to

separate distinct geometric intents. In contrast, GAM (a) reveals compact, well-separated clusters, confirming that SAF induces a Topological Collapse where task-specific manifolds are clearly distinguished.

To further quantify this, we perform Representational Similarity Analysis on the backbone hidden-state representations for PutEggplant task (Fig. 5), where the dataset is constructed such that adjacent indices correspond to the same object placed vertically vs. horizontally. The Baseline RDM (b) shows a diffuse, unstructured distribution: pairwise dissimilarities are scattered without any clear block structure, indicating that the model treats spatially rotated instances inconsistently. Conversely, GAM (a) exhibits a distinct checkerboard structure, where pairs of rotationally-equivalent actions form low-dissimilarity blocks along the off-diagonal. This pattern indicates strong **Spatial Invariance**: the model assigns high similarity to action pairs that differ only in orientation (vertical vs. horizontal), demonstrating that SAF has effectively filtered out the extrinsic pose information ($SE(3)$ group orbits) and aligned the representation with the intrinsic geometric intent. An additional correlation analysis is provided in Sec. A.5.1.

### 4.5. RQ3: Ablation Study

We ablate the two geometric modules (SAF, ALP) and the two training objectives ($\mathcal{L}_{plan}$, $\mathcal{L}_{vel}$) to verify the necessity of each component.

**Effects of Manifold Alignment.** Table 3a shows that SAF and ALP provide complementary benefits. Without either module, performance remains moderate across all tasks, indicating that naive manifold construction offers limited benefit. ALP particularly benefits goal-conditioned variation (Goal 96.8%), while SAF contributes more to long-horizon consistency (Long 89.1%). Their combination yields the best overall balance, especially on Spatial (98.6%) and Long (92.0%), with only a marginal trade-off on Goal compared to ALP alone.

**Effects of Training Objectives.** Table 3b shows that $\mathcal{L}_{plan}$ is indispensable for producing meaningful continuous trajectories: removing it leads to severe collapse across all splits. In contrast, removing $\mathcal{L}_{vel}$ retains reasonably strong performance but consistently underperforms the full objective, achieving 87.3% Long. The results indicate that intent

supervision improves schema selection and long-horizon consistency. Using both losses yields the best results on every split, reaching 98.6% Spatial and 92.0% Long.

## 5. Conclusion

We addressed the fundamental violation of *General Covariance* in robotic learning by proposing the **Generalized Action Manifold (GAM)**. By structurally disentangling Geometric Invariance and Temporal Invariance, GAM induces a Topological Collapse, transforming the intractable global non-convex search into guaranteed local convex refinement. Empirically, this rigorous geometric framework yields better generalization under spatial and temporal distribution shifts, establishing a principled foundation for physically consistent embodied intelligence.

## Impact Statement

This paper advances embodied intelligence by reframing action learning around geometric structure rather than raw parameter scaling. GAM offers an inductive bias that *complements* scale: instead of spending capacity to re-learn invariances that are mathematically free, the model externalizes them into the representation, yielding better trade-offs between data, compute, and out-of-distribution robustness as VLA models and datasets grow. Physically consistent action representations also contribute to safer robotic behavior by mitigating mode-averaging failures. We do not foresee specific negative consequences beyond those generally associated with progress in autonomous physical systems.

## Acknowledgements

This work was supported by the National Natural Science Foundation of China under Grants U23A20387, 62502518, 62532003, U25A20536, in part by the Pengcheng Laboratory Research Project under Grant PCL2023A08, and also in part by the Postdoctoral Fellowship Program of CPSF under Grant Number GZC20251036.

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

# A. Appendix

This appendix provides detailed theoretical proofs, experimental settings, and additional analyses to support the claims made in the main paper. We organize the content as follows:

- **Section A.1: Theoretical Proofs.** We provide detailed derivations for Theorem 1 (Diffeomorphism Invariance) and Theorem 2 (Convexity in Quotient Space), which motivate the temporal and geometric invariance modules in GAM.

- **Section A.2: Implementation Details.** We elaborate on the network architecture, the tokenization pipeline, and the training hyperparameters for reproducibility.

- **Section A.3: Training Pipeline.** We describe the training process, including the data pipeline, training stages, and hyperparameters used to train our model.

- **Section A.4: Dataset Details.** We describe the data collection process, statistics, and task definitions for both real-world and simulation benchmarks.

- **Section A.5: Additional Experiments.** We present extended ablation studies on hyperparameters (*e.g.,* codebook size, weight coefficients) and visualize the learned manifolds.

- **Section A.6: Evaluation Metrics.** We formally define the success metrics and evaluation protocols used in our experiments.

## A.1. Theoretical Proofs

In this section, we provide rigorous mathematical derivations for the theorems presented in the main paper, specifically focusing on the necessity of Diffeomorphism Invariance (Theorem 1) and Quotient Convexity (Theorem 2).

### A.1.1. PROOF OF THEOREM 1: DIFFEOMORPHISM INVARIANCE

**Theorem 1.** *Let $\mathcal{T}$ be the group of monotonic temporal diffeomorphisms. If a representation mapping $\Phi$ satisfies invariance under $\mathcal{T}$ (i.e., $\Phi(\gamma) = \Phi(\gamma \circ \phi)$ for all $\phi \in \mathcal{T}$), then the induced metric $d_\Phi(\gamma_1, \gamma_2) = \|\Phi(\gamma_1) - \Phi(\gamma_2)\|$ assigns zero distance to trajectories identical in geometry but distinct in speed. Under this metric, the regression target for a set of time-warped demonstrations collapses to a Dirac delta function, eliminating temporal variance.*

*Proof.* Let $\gamma(t) \in \mathbb{R}^d$ be the underlying geometric path of an action. In a real-world dataset, we observe time-warped realizations of this path. Let an observed trajectory be denoted as $\tau(t)$, which is subject to a random temporal jitter $\delta(t)$. Formally, we define the time-warped observation as:

$$\tau(t) = \gamma(t + \delta(t)), \tag{12}$$

where $\delta(t)$ is a zero-mean random noise variable representing the deviation from the canonical timeline, with variance:

$$\mathbb{E}[\delta(t)] = 0, \quad \mathbb{E}[\delta(t)^2] = \sigma_t^2. \tag{13}$$

**Part 1: Error Analysis in Ambient Space (Without Invariance).** We analyze the expected error of a direct regression model trained via Mean Squared Error (MSE). The objective is to minimize the Euclidean distance between the model's prediction $\hat{\gamma}(t)$ and the stochastic observation $\tau(t)$. It is a standard result in statistics that the optimal estimator minimizing the MSE is the conditional expectation of the target distribution. Thus, the model converges to:

$$\hat{\gamma}^*(t) = \mathbb{E}_\delta[\tau(t)] = \mathbb{E}_\delta[\gamma(t + \delta(t))]. \tag{14}$$

To quantify the deviation of this estimator from the true geometric path $\gamma(t)$, we perform a first-order Taylor expansion of $\gamma(t + \delta(t))$ around $t$:

$$\gamma(t + \delta(t)) \approx \gamma(t) + \dot{\gamma}(t) \cdot \delta(t), \tag{15}$$

where $\dot{\gamma}(t) = \frac{d\gamma}{dt}$ is the instantaneous velocity vector. The expected approximation error (MSE) is given by:

$$\mathcal{L}_{amb} = \mathbb{E}_\delta \left[ \|\gamma(t + \delta(t)) - \gamma(t)\|^2 \right]. \tag{16}$$

Substituting the Taylor expansion into the error term:

$$\mathcal{L}_{amb} \approx \mathbb{E}_\delta \left[ \|(\gamma(t) + \dot{\gamma}(t)\delta(t)) - \gamma(t)\|^2 \right]. \tag{17}$$

Simplifying the terms inside the norm:

$$\mathcal{L}_{amb} \approx \mathbb{E}_\delta \left[ \|\dot{\gamma}(t)\delta(t)\|^2 \right]. \tag{18}$$

Since $\delta(t)$ is a scalar and $\dot{\gamma}(t)$ is a vector, we can factor out the norms:

$$\mathcal{L}_{amb} \approx \|\dot{\gamma}(t)\|^2 \cdot \mathbb{E}_\delta[\delta(t)^2]. \tag{19}$$

Substituting the variance $\sigma_t^2 = \mathbb{E}[\delta(t)^2]$, we arrive at the error bound:

$$\mathcal{L}_{amb} \approx \|\dot{\gamma}(t)\|^2 \cdot \sigma_t^2. \tag{20}$$

This derivation reveals a critical pathology: the regression error is directly proportional to the squared velocity $\|\dot{\gamma}(t)\|^2$. In high-frequency regions of the trajectory where velocity changes rapidly (e.g., impacts, sudden stops), $\|\dot{\gamma}(t)\|$ is large, causing the error to explode. To minimize this objective, the model is biased towards predicting a trajectory with smaller derivatives ($\|\dot{\gamma}\| \to 0$), effectively acting as a **low-pass filter** that smooths out critical dynamics.

**Part 2: Error Analysis with Temporal Invariance (GAM).** Now, consider the proposed representation mapping $\Phi$ (e.g., Arc-Length Parameterization) that satisfies invariance under the group of temporal diffeomorphisms $\mathcal{T}$. The invariance property implies:

$$\Phi(\gamma(t)) = \Phi(\gamma(t + \delta(t))), \quad \forall \delta(t) \text{ s.t. } (t + \delta) \text{ is monotonic.} \tag{21}$$

Let $s = \Phi(\tau)$ be the target representation in the invariant space. The sensitivity of this target to the temporal jitter $\delta$ is given by the partial derivative:

$$\frac{\partial \Phi(\gamma(t + \delta))}{\partial \delta} = 0. \tag{22}$$

We now re-evaluate the expected error in this invariant feature space. The first-order Taylor expansion of the error is:

$$\mathcal{L}_{inv} \approx \mathbb{E}_\delta \left[ \left\| \frac{\partial \Phi(\gamma(t))}{\partial \delta} \cdot \delta(t) \right\|^2 \right]. \tag{23}$$

Since the derivative is identically zero, the error term vanishes:

$$\mathcal{L}_{inv} = \mathbb{E}_\delta[\|\mathbf{0} \cdot \delta(t)\|^2] = 0. \tag{24}$$

This implies that the variance of the target representation $s$ conditioned on the geometric path $\gamma$ is zero:

$$\mathrm{Var}(s|\gamma) = 0. \tag{25}$$

Consequently, the conditional probability density function of the target collapses to a single point mass:

$$P(s|\gamma) = \delta_{\text{Dirac}}(s - \Phi(\gamma)), \tag{26}$$

where $\delta_{\text{Dirac}}$ denotes the Dirac delta function. **Conclusion:** By enforcing temporal invariance, the irreducible error caused by misalignment is eliminated. Unlike direct regression, the error bound becomes completely decoupled from the signal velocity $\|\dot{\gamma}(t)\|$, allowing the model to fit high-frequency geometric details without being forced to apply low-pass smoothing. $\square$

### A.1.2. PROOF OF THEOREM 2: CONVEXITY IN QUOTIENT SPACE

**Theorem 2.** *Let $\mathcal{M}$ be the action manifold invariant under the joint symmetry group $\mathcal{G}$. The optimization problem is multi-valued in the ambient space because the set of orbits is not closed under linear addition. However, by projecting the problem onto the Quotient Space $\mathcal{Q} = \mathcal{M}/\mathcal{G}$, the optimization landscape is simplified. We prove that for any two trajectories $\mathbf{a}_1, \mathbf{a}_2 \in \mathcal{M}$ belonging to the same geometric schema in $\mathcal{Q}$, their geodesic interpolation remains valid within $\mathcal{Q}$, whereas their linear interpolation in the ambient space exits $\mathcal{M}$.*

*Proof.* Let the Generalized Action Manifold be defined as the union of group orbits: $\mathcal{M} = \bigcup_{h \in \mathcal{G}} h \cdot \mathcal{S}$, where $\mathcal{S}$ is the base shape space and $\mathcal{G}$ is the symmetry group (including spatial $SE(3)$ transforms). Consider two trajectories $\mathbf{a}_1, \mathbf{a}_2 \in \mathcal{M}$ that belong to the same geometric schema, meaning they share the same intrinsic canonical shape $\mathbf{x}_c \in \mathcal{S}$ but lie on different orbits defined by group elements $g_1, g_2 \in \mathcal{G}$. Assuming a left group action, we write:

$$\mathbf{a}_1 = g_1 \cdot \mathbf{x}_c, \quad \mathbf{a}_2 = g_2 \cdot \mathbf{x}_c. \tag{27}$$

**Part 1: Non-Convexity in Ambient Space.** The standard regression objective inherently minimizes the Euclidean distance. The optimal solution lies at the arithmetic mean of the demonstrations. Let $\bar{\mathbf{a}}$ be the linear convex combination of the two trajectories with weight $\alpha \in (0, 1)$:

$$\bar{\mathbf{a}} = \alpha \mathbf{a}_1 + (1 - \alpha)\mathbf{a}_2 = (\alpha g_1 + (1 - \alpha)g_2) \cdot \mathbf{x}_c. \tag{28}$$

The term $\bar{g} = \alpha g_1 + (1 - \alpha)g_2$ represents the linear interpolation of the group elements. We examine whether $\bar{g}$ remains a valid element of the symmetry group $\mathcal{G}$. Let us focus on the spatial rotation component $R \in SO(3)$, which is a subgroup of $\mathcal{G}$. The defining property of $SO(3)$ is orthogonality: $R^\top R = I$. Substituting the interpolated matrix $\bar{R} = \alpha R_1 + (1 - \alpha)R_2$, we compute the product $\bar{R}^\top \bar{R}$:

$$\bar{R}^\top \bar{R} = (\alpha R_1 + (1 - \alpha)R_2)^\top (\alpha R_1 + (1 - \alpha)R_2). \tag{29}$$

Expanding this term:

$$\bar{R}^\top \bar{R} = \alpha^2 R_1^\top R_1 + (1 - \alpha)^2 R_2^\top R_2 + \alpha(1 - \alpha)(R_1^\top R_2 + R_2^\top R_1). \tag{30}$$

Using the identity $R_i^\top R_i = I$:

$$\bar{R}^\top \bar{R} = (\alpha^2 + (1 - \alpha)^2)I + \alpha(1 - \alpha)(R_1^\top R_2 + R_2^\top R_1). \tag{31}$$

For generic $R_1 \neq R_2 \in SO(3)$ and $\alpha \in (0, 1)$, $\bar{R}^\top \bar{R} \neq I$, so $\bar{R}$ does not preserve orthogonality and hence $\bar{R} \notin SO(3)$ except in degenerate cases (*e.g.*, $R_1 = R_2$, or trivial $\alpha \in \{0, 1\}$). Therefore:

$$\bar{g} \notin \mathcal{G} \implies \bar{\mathbf{a}} \notin \mathcal{M}. \tag{32}$$

Since the Euclidean mean $\bar{\mathbf{a}}$ does not lie on any valid group orbit, it falls into the "forbidden zone" off the manifold. Minimizing the distance to this invalid point creates a saddle point in the optimization landscape, leading to mode averaging.

**Part 2: Convexity in Quotient Space.** In contrast, the GAM framework maps actions to the quotient space $\mathcal{Q} = \mathcal{M}/\mathcal{G}$ via the canonical projection $\mathcal{P} : \mathcal{M} \to \mathcal{Q}$. This operator extracts the intrinsic shape by factoring out the group action:

$$\mathcal{P}(\mathbf{a}) = \inf_{g \in \mathcal{G}} \|\mathbf{a} - g \cdot \mathbf{x}_c\| \implies \mathcal{P}(\mathbf{a}) \equiv \mathbf{x}_c. \tag{33}$$

Applying this to our two trajectories:

$$\mathcal{P}(\mathbf{a}_1) = \mathbf{x}_c, \quad \mathcal{P}(\mathbf{a}_2) = \mathbf{x}_c. \tag{34}$$

The projection factors out the conflicting group orbits $g_1, g_2$ perfectly, yielding the identical canonical shape. The optimization objective in the quotient space $\mathcal{Q}$ collapses to minimizing the distance to this single, consistent target $\mathbf{x}_c$:

$$\mathcal{L}_\mathcal{Q}(\hat{\mathbf{x}}) = \|\hat{\mathbf{x}} - \mathbf{x}_c\|^2. \tag{35}$$

We analyze the Hessian of this loss function with respect to the prediction $\hat{\mathbf{x}}$:

$$\nabla^2_{\hat{\mathbf{x}}} \mathcal{L}_\mathcal{Q} = \frac{\partial^2}{\partial \hat{\mathbf{x}}^2} \left( (\hat{\mathbf{x}} - \mathbf{x}_c)^\top (\hat{\mathbf{x}} - \mathbf{x}_c) \right) = 2\mathbf{I}. \tag{36}$$

Since the identity matrix $\mathbf{I}$ is strictly positive definite ($2\mathbf{I} \succ 0$), the loss function $\mathcal{L}_\mathcal{Q}$ is strictly convex. **Conclusion:** By projecting the problem into the quotient space, the multi-modal optimization landscape (characterized by saddle points) is transformed into a unimodal convex basin. This guarantees a unique global optimum and ensures convergence to a valid geometric intent without mode-averaging artifacts. □

## A.2. Implementation Details

We implement the GAM-VLA framework using a hierarchical *planner-Executor* architecture, where a Vision-Language Model (VLM) serves as the high-level geometric planner and a Conditional Flow Matching network acts as the low-level geometric executor. This section details the network specifications, the derivation of supervision signals, and the precise training objectives.

### A.2.1. NETWORK ARCHITECTURE

**VLA Planner (Backbone).** We initialize the planner policy $\pi_\theta$ with the pre-trained **Qwen2.5-VL-3B-Instruct** model. The visual encoder is a SigLIP-based Vision Transformer that processes RGB images into unpooled patch embeddings, preserving spatial granularity. The language tokens are processed by the LLM decoder with a hidden dimension of $D = 2048$. To adapt this backbone for robotic control, we append three specific prediction heads to the final transformer layer. First, the **Schema Head** is a linear classifier projecting the hidden state to the codebook size $K = 1024$ to predict the discrete geometric intent $z_s$. Second, the **Modulation Head** is a Multi-Layer Perceptron (MLP) that regresses the continuous affine parameters $z_{pose} \in \mathbb{R}^{10}$ (comprising translation $\mu \in \mathbb{R}^3$, rotation $U \in \mathbb{R}^6$ in continuous representation, and scale $\sigma \in \mathbb{R}^1$). Third, the **Dynamics Head** regresses the scalar temporal factor $z_{time} \in \mathbb{R}^1$.

**Flow Executor (Condition Flow Matching).** For the continuous refinement phase, we employ a Diffusion Transformer (DiT-B) architecture consisting of 16 transformer blocks with a hidden dimension of 1024 and 16 attention heads. A critical design choice is the conditioning mechanism. Unlike standard concatenation, we inject the semantic condition into the DiT via **Adaptive Layer Normalization (AdaLN)**. Let $\mathbf{h}_{intent} \in \mathbb{R}^D$ be the hidden state extracted from the VLA backbone corresponding to the predicted schema token. The AdaLN layer modulates the normalized features $x$ based on the diffusion time step $t$ and the intent condition $\mathbf{h}_{intent}$:

$$\text{AdaLN}(x, t, \mathbf{h}_{intent}) = \gamma(t, \mathbf{h}_{intent}) \odot \text{LayerNorm}(x) + \beta(t, \mathbf{h}_{intent}), \tag{37}$$

where $\gamma$ and $\beta$ are learnable affine projections. This ensures that the generation of the canonical shape $\mathbf{x}_c$ is strictly governed by the high-level geometric intent locked by the planner.

### A.2.2. SUPERVISION AND TRAINING OBJECTIVES

The training follows a decoupled two-stage paradigm: (1) Unsupervised auto-labeling via the Tokenizer, and (2) Supervised learning of the VLA policy.

**Derivation of Ground Truth (Auto-Labeling).** The supervision targets for the VLA are not manually annotated but are derived analytically by the **GAM Tokenizer**. We pre-train the tokenizer on the mixture dataset to learn the geometric codebook $\mathcal{C}$. During VLA training, we freeze the tokenizer and process every raw ground-truth trajectory $\mathbf{a}_{gt}$ to extract a tuple of targets: the discrete schema index $z_s^{gt}$, the analytical affine parameters $z_{pose}^{gt}$, the cumulative arc-length $z_{time}^{gt}$, and the normalized canonical shape $\mathbf{x}_c^{gt}$. These derived labels serve as the ground truth for the subsequent optimization.

**Optimization of the Planner.** The VLA backbone is trained to map visual-linguistic inputs $(\mathbf{o}, \mathbf{l})$ to these tokenizer-derived targets. We optimize a multi-task planning objective $\mathcal{L}_{\text{plan}}$ that combines classification and regression losses:

$$\mathcal{L}_{\text{plan}} = \lambda_{cls} \underbrace{\mathcal{L}_{\text{CE}}(\hat{z}_s, z_s^{gt})}_{\text{Intent Locking}} + \lambda_{reg} \left( \|\hat{z}_{pose} - z_{pose}^{gt}\|^2 + \|\hat{z}_{time} - z_{time}^{gt}\|^2 \right). \tag{38}$$

Here, $\mathcal{L}_{\text{CE}}$ is the cross-entropy loss that forces the model to select the correct solution basin, while the Mean Squared Error (MSE) terms ensure accurate spatial grounding and speed estimation. We apply Low-Rank Adaptation (LoRA) to the attention weights of the backbone to prevent catastrophic forgetting of pre-trained knowledge.

**Optimization of the Executor.** The Flow Head is trained to generate the prediction-conditioned, arc-length-aligned trajectory $\boldsymbol{\xi}_{gt}^{alp}$ starting from Gaussian noise $\mathbf{x}_0 \sim \mathcal{N}(\mathbf{0}, \mathbf{I})$. We adopt the Optimal Transport path $\psi_t(\mathbf{x}_0) = (1 - t)\mathbf{x}_0 + t\boldsymbol{\xi}_{gt}^{alp}$. The network $v_\phi$ is trained to predict the vector field $u_t = \boldsymbol{\xi}_{gt}^{alp} - \mathbf{x}_0$ by minimizing the executor objective, which combines flow matching with the velocity regularization $\mathcal{L}_{\text{vel}}$:

$$\mathcal{L}_{\text{exec}} = \mathbb{E}_{t, \mathbf{x}_0} \left[ \|v_\phi(\psi_t(\mathbf{x}_0), t, \mathbf{h}_{intent}) - (\boldsymbol{\xi}_{gt}^{alp} - \mathbf{x}_0)\|^2 \right] + \lambda \mathcal{L}_{\text{vel}}. \tag{39}$$

---

**Algorithm 1** GAM-VLA Training Pipeline

---

1: **Input:** Dataset $\mathcal{D} = \{\gamma_i\}$, VLA Policy $\pi_\theta$, Flow Net $v_\phi$
2: *// Stage 1: Unsupervised Tokenization*
3: **for** each trajectory $\gamma \in \mathcal{D}$ **do**
4:     Extract affine parameters $(\mu, U, \sigma)$ via SAF; Compute canonical shape $\mathbf{x}_c$.
5: **end for**
6: Train RVQ-VAE on $\{\mathbf{x}_c\}$ to obtain codebook $\{\mathbf{e}_k\}$ and labels $\{z_s^{gt}\}$.
7: *// Stage 2: Structured Policy Learning*
8: **while** not converged **do**
9:     Sample batch $(\mathbf{o}, \mathbf{l}, z_s^{gt}, \mathbf{x}_c^{gt}) \sim \mathcal{D}$.
10:     **Planner Forward:** $\hat{z}_s, \mathbf{h}_{\text{intent}} = \pi_\theta(\mathbf{o}, \mathbf{l})$.
11:     Compute Planner Loss: $\mathcal{L}_{\text{plan}}$ (Eq. 9).
12:     **Executor Forward:** Sample $t, \mathbf{x}_0$; Predict vector field $\hat{v} = v_\phi(\psi_t, t, \mathbf{h}_{\text{intent}})$.
13:     Compute Executor Loss: $\mathcal{L}_{\text{exec}}$ (Eq. 10).
14:     Update $\theta, \phi$ to minimize $\mathcal{L}_{\text{plan}} + \mathcal{L}_{\text{exec}}$.
15: **end while**

---

Crucially, the condition $\mathbf{h}_{intent}$ is obtained from the VLA backbone in a teacher-forcing manner using the ground-truth schema token, ensuring the flow head learns to generate geometry consistent with valid geometric intents. The final total loss is the summation $\mathcal{L}_{\text{total}} = \mathcal{L}_{\text{plan}} + \mathcal{L}_{\text{exec}}$.

### A.2.3. TRAINING CONFIGURATION

We summarize the detailed hyperparameters for the model architecture, optimization, and diffusion process in Table 5.

*Table 5.* **Comprehensive Hyperparameters for GAM-VLA Training.**

| Category | Parameters and Values |
|---|---|
| **Architecture** | |
| VLM Backbone | Qwen2.5-VL-3B-Instruct |
| Vision Backbone | Internal SigLIP-based ViT (unpooled features) |
| VLM Hidden Dimension | 2048 |
| Tokenizer Structure | 3-layer RVQ (Codebook $K = 1024$, Dim $D = 256$) |
| Action Model Type | DiT-B (16 layers, 1024 dim, 16 heads) |
| Conditioning | Cross-Attention via Adaptive LayerNorm (ada_norm) |
| **VLA Training** | |
| Max Training Steps | 100,000 |
| Warmup Steps | 5,000 (Ratio: 0.05) |
| Base Learning Rate | $1.0 \times 10^{-5}$ (VLM LoRA) / $1.0 \times 10^{-4}$ (DiT) |
| LR Scheduler | Cosine Decay with Min LR ($5.0 \times 10^{-7}$) |
| Optimizer | AdamW ($\beta_1 = 0.9, \beta_2 = 0.95, \epsilon = 10^{-8}$) |
| Weight Decay | 0.1 |
| Precision | Mixed Precision (BF16) |
| **Action & Diffusion** | |
| Action Space | Pose-Normalized Canonical Frame |
| Horizon ($T$) / Dim ($D$) | 30 / 6 (Pos+Rot) + 1 (Gripper) |
| Diffusion Steps (Train) | Continuous ($t \sim \mathcal{U}[0, 1]$) |
| Diffusion Steps (Inference) | 10 steps (via Euler ODE Solver) |

### A.3. Training Pipeline for GAM-VLA

In this subsection, we describe the training pipeline used for the Generalized Action Manifold (GAM) in the VLA framework. The training consists of two main stages: unsupervised tokenization and structured policy learning. These stages are essential for generating effective action representations and training the VLA policy.

**Stage 1: Unsupervised Tokenization.** In this stage, we extract canonical shapes from input trajectories and train an

RVQ-VAE on the resulting shape distribution to obtain a codebook of action prototypes, $\mathbf{e}_k$, and their corresponding labels, $z_s$. These action prototypes serve as discrete action intents for the subsequent structured learning stage.

**Stage 2: Structured Policy Learning.** In this stage, the VLA model is trained by sampling from the dataset and performing forward passes through both the planner and the executor. The planner predicts discrete intent labels, $z_s$, and the executor generates the continuous action trajectory, $\hat{v}$. The training is guided by two losses: the Planner Loss ($\mathcal{L}_{plan}$) and the Executor Loss ($\mathcal{L}_{exec}$). The policy parameters $\theta$ and $\phi$ are updated to minimize these losses, thereby improving both the intent prediction and the continuous action generation.

### A.4. Dataset Details

#### A.4.1. SIMULATION BENCHMARKS

**LIBERO.** To evaluate the versatile manipulation capabilities of our model, we utilize the *LIBERO* (Lifelong Robot Learning) benchmark, which consists of four distinct task suites designed to test different facets of generalization. The **LIBERO-Spatial** suite contains 10 tasks that require the robot to reason about spatial relationships (e.g., relative positioning). The **LIBERO-Object** suite focuses on visual generalization across different object textures and geometries. The **LIBERO-Goal** suite tests the agent's ability to discern distinct instructions that involve identical scenes. Most critically for our framework, the **LIBERO-Long** suite consists of long-horizon tasks composed of sequential sub-goals. We utilize the standard dataset provided by the benchmark, which includes 50 human demonstrations for each of the 10 tasks per suite. We treat LIBERO-Long as the primary proxy for evaluating the *Topological Collapse* and temporal consistency of the constructed action manifold.

**SimplerEnv.** To rigorously assess the robustness of our General Covariant policies against distribution shifts, we employ the *SimplerEnv* benchmark on the WidowX robot embodiment. Unlike standard simulation environments with static settings, SimplerEnv is designed to replicate real-world visual complexities. We evaluate our model on four manipulation tasks: *Put Spoon*, *Put Carrot*, *Stack Block*, and *Put Eggplant*. For each task, the evaluation is conducted under three distinct distribution shift settings: (1) **Visual Shift**, where the background textures and lighting conditions are randomized; (2) **Camera Shift**, where the camera pose is perturbed to test viewpoint invariance; and (3) **Dynamics Shift**, where the physical properties of objects are altered. These settings specifically test whether the GAM framework has successfully decoupled the invariant task geometry from the equivariant environmental parameters.

#### A.4.2. CONFIGURATION FOR REPRESENTATION ANALYSIS

To support the empirical analysis in RQ1 and RQ2, we constructed specific evaluation subsets to visualize the learned manifold structure.

**Setup for t-SNE Visualization.** We aim to visualize whether the model's hidden states cluster according to task semantics (Geometric Intent) rather than low-level trajectory variations. We constructed a multi-task analysis set sampling evaluation episodes from four distinct SimplerEnv-WidowX tasks: *Put Carrot on Plate*, *Put Eggplant into Yellow Basket*, *Put Spoon on Towel*, and *Stack Green Block on Yellow Block*. For each episode, we extract the **Action Intent Hidden State** $\mathbf{h}_{intent}$ from the VLA backbone. We use t-Distributed Stochastic Neighbor Embedding (t-SNE) with a perplexity of 30 and a learning rate of 200 to project these high-dimensional hidden states into a 3D space. The resulting clusters are colored by ground-truth task labels to verify if the representation induces a geometric collapse where functionally similar actions group together.

**Setup for Representational Similarity Analysis (RSA).** To quantify the geometric alignment between the latent space and the underlying action geometry, we focus on the *Put Eggplant* task from SimplerEnv, which exhibits strong rotational symmetry. We collected a specialized validation set in which adjacent indices correspond to the same object placed in vertical vs. horizontal poses, so that ideal spatial invariance would manifest as a low-dissimilarity block structure. For each model, we compute the **Hidden State Representational Dissimilarity Matrix (RDM)** using the cosine distance between the model's latent vectors $\mathbf{h}_{intent}$:

$$M_{i,j}^{lat} = 1 - \frac{\mathbf{h}_i^\top \mathbf{h}_j}{\|\mathbf{h}_i\|\|\mathbf{h}_j\|}. \tag{40}$$

Figure 5 visualizes the resulting Hidden State RDMs for GAM and the Baseline. For the quantitative analysis in Sec. A.5.1, we further compute an **Action Dissimilarity** signal using the Euclidean distance between the ground-truth canonical shapes $\mathbf{x}_c$ (after affine disentanglement), and report the Spearman rank correlation $\rho$ between the latent dissimilarities and action dissimilarities. A high $\rho$ indicates that the rank-order structure of distances in the latent space is isomorphic to that of the

intrinsic action geometry, providing a quantitative measure of General Covariance.

## A.5. Additional Experiments and Analysis

### A.5.1. MANIFOLD ALIGNMENT QUANTIFICATION VIA RSA SCATTER PLOTS

To deeply investigate the geometric alignment quality, we visualize the correlation between the *Action Dissimilarity* (ground truth distance) and *Hidden State Dissimilarity* (latent distance) in Figure 6.

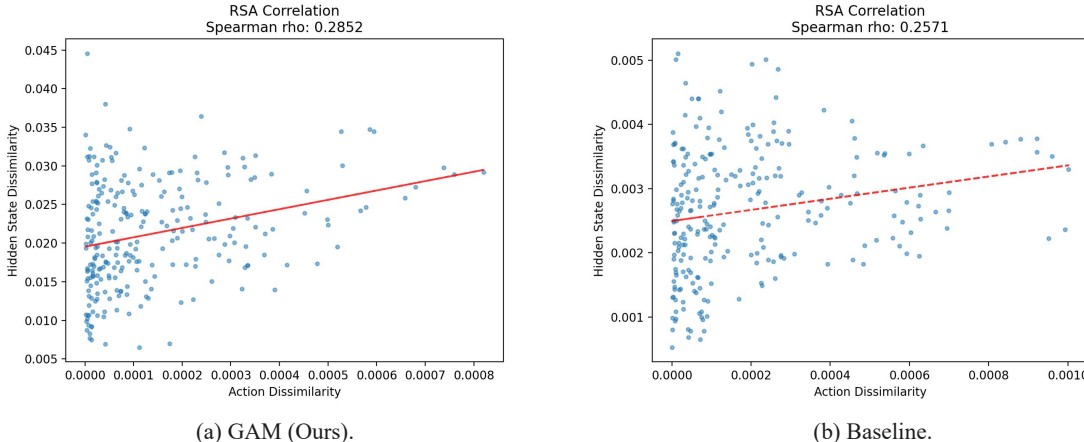

(a) GAM (Ours).      (b) Baseline.

*Figure 6.* **RSA Scatter Plots.** Comparison of the representational alignment between GAM (a) and Baseline (b). The x-axis represents the pairwise Euclidean distance between ground-truth canonical actions, and the y-axis represents the cosine distance between the corresponding hidden states.

As shown in Fig. 6, GAM (a) yields a slightly higher Spearman rank correlation ($\rho = 0.2852$) than the Baseline ($\rho = 0.2571$), suggesting improved local topology preservation. Although the absolute gain is moderate, the low-dissimilarity region (left side of the x-axis) exhibits more compact latent distances for geometrically similar actions, indicating that our model maps such actions to nearby points on the latent manifold. In contrast, the Baseline (b) shows higher variance in latent distances even for similar actions, suggesting a more entangled representation space.

### A.5.2. HYPERPARAMETER SENSITIVITY

We conduct sensitivity analysis on key hyperparameters to understand their impact on manifold construction and policy performance. The results are summarized in Table 6.

**1. Impact of Velocity Regularization Weight $\lambda$.** As shown in Table 6a, the weight $\lambda$ is critical for preventing stationary solutions. With a low $\lambda = 0.1$, the model prioritizes pure geometric fit but fails to output sufficient velocity, resulting in a 34.4% success rate. Conversely, a high $\lambda = 10.0$ causes the model to overfit to noisy velocity profiles, leading to degraded geometric precision (45.9%). The optimal $\lambda = 2.0$ effectively balances geometric fidelity and temporal consistency, resulting in a 56.3% average success rate.

**2. Impact of Codebook Size $K$.** Table 6b demonstrates the effect of schema granularity. A small codebook ($K = 256$) forces disparate actions into the same cluster (mode collapse), leading to under-fitting and a lower average success rate (42.7%). A very large codebook ($K = 4096$) results in sparse clusters, causing instability in covariance estimation and whitening (52.1%). The optimal $K = 1024$ achieves the best balance, providing the highest average success rate of 56.3%.

**3. Impact of Latent Commitment Weight $\beta$.** Table 6c highlights the importance of $\beta$ in balancing latent commitment and reconstruction error. With a low $\beta = 0.1$, the model struggles with latent representation, causing degradation on tasks like Stack Block (37.5%) and Put Eggplant (75.0%). A high $\beta = 10.0$ overemphasizes commitment, reducing performance on Stack Block (33.3%) and Put Eggplant (83.3%). The optimal $\beta = 1.0$ provides the best trade-off, achieving consistent performance across all tasks (*e.g.,* Stack Block 41.7%, Put Eggplant 87.5%).

*Table 6.* **Hyperparameter Sensitivity Analysis on SimplerEnv.** We report Success Rates (SR) across four tasks. The default settings ($\lambda = 2.0, K = 1024, N = 16$) are highlighted in blue.

*(a)* **Impact of Velocity Regularization Weight $\lambda$.** $\lambda$ balances geometric fidelity and temporal consistency.

| $\lambda$ | Put Spoon | Put Carrot | Stack Block | Put Eggplant | Avg. |
|---|---|---|---|---|---|
| 0.1 (Low) | 33.3% | 29.2% | 12.5% | 62.5% | 34.4% |
| **2.0 (Optimal)** | **58.3%** | **37.5%** | **41.7%** | **87.5%** | **56.3%** |
| 10.0 (High) | 50.0% | 37.5% | 16.7% | 79.2% | 45.9% |

*(b)* **Impact of Schema Codebook Size $K$.** $K$ determines the granularity of geometric clustering.

| Codebook $K$ | Put Spoon | Put Carrot | Stack Block | Put Eggplant | Avg. |
|---|---|---|---|---|---|
| 256 (Small) | 50.0% | 33.3% | 20.8% | 66.7% | 42.7% |
| **1024 (Optimal)** | **58.3%** | **37.5%** | **41.7%** | **87.5%** | **56.3%** |
| 4096 (Large) | 54.2% | 37.5% | 29.2% | 87.5% | 52.1% |

*(c)* **Impact of $\beta$ on Latent Commitment.** $\beta$ controls the trade-off between reconstruction error and latent commitment.

| $\beta$ | Put Spoon | Put Carrot | Stack Block | Put Eggplant | Avg. |
|---|---|---|---|---|---|
| 0.1 (Low) | 58.3% | 41.7% | 37.5% | 75.0% | 53.1% |
| **1.0 (Optimal)** | **58.3%** | **37.5%** | **41.7%** | **87.5%** | **56.3%** |
| 10.0 (High) | 50.0% | 45.8% | 33.3% | 83.3% | 53.1% |

## A.6. Evaluation Metrics

To comprehensively assess both the downstream control performance and the intrinsic quality of the learned representations, we utilize a diverse set of metrics ranging from task success rates to geometric alignment scores.

**Task Performance Metrics.** The primary metric for policy evaluation is the **Success Rate (SR)**, defined as the percentage of evaluation episodes where the agent successfully satisfies the goal condition within the maximum horizon. For a set of $N$ evaluation trials, $SR = \frac{1}{N} \sum_{i=1}^{N} \mathbb{I}(\text{success}_i)$. In simulation benchmarks (LIBERO, SimplerEnv), success is determined by the environment's ground-truth state detector. To quantify the fidelity of the tokenizer, we report the **Mean Absolute Error (MAE)** between the ground truth trajectory $\mathbf{a}$ and the reconstructed trajectory $\hat{\mathbf{a}}$. For a dataset $\mathcal{D}$, this is calculated as $\text{MAE} = \frac{1}{|\mathcal{D}| \cdot T} \sum_{\mathbf{a} \in \mathcal{D}} \sum_{t=1}^{T} \|\mathbf{a}_t - \hat{\mathbf{a}}_t\|_1$. Additionally, the **Compression Ratio (CR)** measures the efficiency of the representation, defined as the ratio of the original raw dimensionality ($T \times D_{action}$) to the number of discrete tokens used by the model.

**Manifold Structure Analysis (t-SNE).** To visualize the geometric structure of the learned latent space, we employ **t-Distributed Stochastic Neighbor Embedding (t-SNE)**. We extract the action intent hidden states $\mathbf{h}_{intent} \in \mathbb{R}^d$ from the VLA backbone for a subset of validation episodes. We project these high-dimensional vectors into $\mathbb{R}^3$ by minimizing the Kullback-Leibler divergence between the joint probabilities of the low-dimensional embedding and the high-dimensional data. This metric qualitatively validates the *Topological Collapse*: if GAM works as intended, embeddings should form compact clusters based on action schemas (e.g., "grasping") regardless of variations in spatial position or orientation.

**Manifold Alignment Quantification (RSA).** To quantitatively evaluate the alignment between the learned semantic geometry and the physical action manifold, we employ **Representational Similarity Analysis (RSA)**. We construct two Representational Dissimilarity Matrices (RDMs). First, the **Latent RDM** $\mathbf{M}_{latent} \in \mathbb{R}^{N \times N}$ computes the pairwise cosine distances between the model's hidden states:

$$M_{i,j}^{lat} = 1 - \frac{\mathbf{h}_i^\top \mathbf{h}_j}{\|\mathbf{h}_i\| \|\mathbf{h}_j\|}. \tag{41}$$

Second, the **Physical RDM** $\mathbf{M}_{phys} \in \mathbb{R}^{N \times N}$ computes the pairwise Euclidean distances between the ground-truth canonical shapes (invariant geometry):

$$M_{i,j}^{phys} = \|\mathbf{x}_c^{(i)} - \mathbf{x}_c^{(j)}\|_2. \tag{42}$$

To compute the alignment score, we extract the upper triangular elements of both matrices into vectors $\mathbf{v}_{lat}$ and $\mathbf{v}_{phys}$. The final metric $\rho$ is the **Spearman rank correlation**, defined as the Pearson correlation coefficient between the ranked variables:

$$\rho = \frac{\text{cov}(\mathbf{r}_{lat}, \mathbf{r}_{phys})}{\sigma_{\mathbf{r}_{lat}} \sigma_{\mathbf{r}_{phys}}}, \quad \text{where } \mathbf{r} = \text{rank}(\mathbf{v}). \tag{43}$$

A high $\rho$ indicates that the rank-order structure of distances in the latent space is isomorphic to that of the intrinsic action geometry, providing a quantitative measure of General Covariance.

