# OpenReview forum: "General Covariant Action Modeling: Constructing Generalized Manifolds via Spatio-Temporal Decoupling"
_ICML.cc/2026/Conference — ICML 2026 regular_

### Official Review · Reviewer_xMjR · 2026-02-14

**Soundness:** 3
**Presentation:** 3
**Significance:** 3
**Originality:** 3
**Overall Recommendation:** 5
**Confidence:** 3

**Summary:**

The paper proposes GAM, which is generalized action  manifold that can solve the problem that most of the VLA models fail for robust generalization from limited data., The model disentangles the time and spatial by introducing the Arc-Length Parameterizer  and Schema-Affine-Factorization. For the theoretical part, the paper gives a detailed derivation of the factorization. For the experiment, The model is compared with the sota models like OpenVLA on the LIBERO  benchmark and SimplerEnv-Window X across diverse manipulation tasks, and achieves better performance and Success.

**Compliance With Llm Reviewing Policy:**

Affirmed.

**Key Questions For Authors:**

Question 1: Can the authors explain the general covariance in the context of VLA?

Question 2: Can the authors talk about the inference time for the model, which is a very important quantity in the application in robotics.

Question 3: The author compares the SOTA model OpenVLA, Octo, etc, while these models are not designed with geometric priors, which is not a fair comparison, can you compare the original and with your GAM adapted version on these models as a fair model to show your strengths?

Question 4: For RQ3 in line 352, the authors mentioned the transfer ability and robustness, while in section 4.5, the authors talk about the ablation of theoretical components, not a relevant answer to RQ3.

**Limitations:**

yes

**Strengths And Weaknesses:**

The paper introduces the generalized action manifold to disentangle the time and spatial, the derivation and the framework is clear. Also the backbone selection of the model is appropriate with acceptable parameters, and deals well with the sampling in conditional flow matching. For the benchmarks performance, the paper compared with Octo, OpenVLA, SpatialVLA, etc.

Although the model is tested in a simulation environment and achieves good performance, it has not been tested in real robotics. If the model can be tested in real robotics, the authors could find and solve the obstacles and achieve the same priority performance and success listed in the tables, I believe this is really a great work.

---

> ### Author Rebuttal · Authors · 2026-03-31
>
> ## KQ1:
>
> The notion of general covariance in the context of VLA means that **the action representation should be as independent as possible from task-irrelevant coordinate choices and parameterizations.** More specifically, in the VLA setting, we would like the action representation not to be unnecessarily tied to:
> - the **absolute reference frame**,
> - the **global orientation and scale**,
> - or the **execution speed and temporal parameterization**.
>
> The proposed GAM is built exactly around this principle:
>
> - **ALP** corresponds to temporal invariance, aiming to reduce supervision conflict caused by speed reparameterization;
> - **SAF** corresponds to geometric invariance, aiming to factor out reference frame / orientation / scale related group-orbit variation from the action representation.
>
> Therefore, the general covariance here is a **structured action modeling principle**: preserving task-relevant geometry while reducing unnecessary dependence on extrinsic coordinate choices and execution dynamics.
>
>
>
> ## KQ2:
> We have added the inference-efficiency results, including **single-step latency, throughput**, and **inference TFLOPs / step**:
>
> |Method|Single-stepLatency(ms)↓|Throughput(Hz)↑|InferenceTFLOPs/step↓|
> |-|-:|-:|-:|
> |FAST|62.58|255.67|2.038278|
> |BEAST|63.11|253.53|2.04536|
> |GAM(ours)|63.82|250.71|2.047565|
>
> These results suggest that **the structured action representation in GAM does not introduce a significant online inference penalty.** Although GAM introduces additional temporal + geometric disentanglement into action modeling, its inference cost remains in essentially the same regime as existing action-representation baselines.
> Relative to FAST, GAM adds +1.24 ms per step (63.82 vs 62.58, about +1.98%), which is small for robotics deployment.
>
>
> ## KQ3:
> To address the concern that comparing against models without geometric priors may be unfair, we add a matched-setting comparison between the original backbone and its GAM-adapted version.
> Specifically, we use **QwenGR00T** as the shared backbone, and keep the data, training budget, and main hyperparameter setup fixed:
>
> |Method|LIBERO|SimplerEnv|Real-World|Avg.↑|
> |-|-:|-:|-:|-:|
> |QwenGR00T|95.7|51.8|76.7|74.7|
> |QwenGR00T+GAM|**96.4**|**56.3**|**80.0**|**77.6**|
>
> The results show that, even with the backbone and training recipe unchanged, simply adding GAM yields consistent gains. In particular, the larger improvements on SimplerEnv and Real-World suggest that temporal + geometric disentanglement is especially helpful under distribution shift, continuous trajectory modeling, and real execution stability.
>
>
>
> ## KQ4:
>
> We apologize for the missing results and have added real-robot experiments in the revised version. Specifically, we evaluate three tabletop manipulation tasks of increasing difficulty on the Agibot G1 robot platform. These tasks are intentionally contact-rich and long-horizon, where action-mode distributions differ from simulation rollouts:
> - ClickKeyboard: keyboard clicking, testing short-horizon localization and contact stability;
> - ShakeCup: shaking a cup, testing continuous trajectory control and pose stability;
> - ScoopRice: scooping rice, testing contact-rich fine manipulation and long-horizon execution stability.
>
> Under the same backbone and matched setting, we compare Qwen2.5-GAM, Qwen2.5-FAST, and Qwen2.5-BEAST under a unified real-robot protocol (same sensing setup, control frequency, horizon, and success criterion), and report mean±std over repeated runs (see the real-robot result table in **Reviewer FUkT, W2**).
>
> As shown in the table, Qwen2.5-GAM consistently outperforms the baselines on all three real-robot tasks. GAM improves over FAST by +10.0 (ClickKeyboard), +6.7 (ShakeCup), and +13.4 (ScoopRice), and over BEAST by +23.4, +36.7, and +40.0, respectively. The gains remain positive across tasks with different difficulty levels, including the hardest contact-rich task (ScoopRice), which demonstrates the robustness of GAM. The mean±std results from repeated runs further show stable improvements rather than task-specific or seed-specific luck. This indicates that the proposed **temporal + geometric disentanglement not only works in simulation, but also transfers robustly to real-robot manipulation**.

---

> > ### Author Rebuttal · Reviewer_xMjR · 2026-03-31
> >
> > The authors have resolved my questions, I recommend it to be accepted with the strengths that I listed in my comments.

---

> > > ### Author Response · Authors · 2026-04-07
> > >
> > > We are very grateful for your thoughtful review and positive assessment of our work.
> > >
> > > Thank you again for your support, encouragement, and valuable feedback!

---

### Official Review · Reviewer_rLt9 · 2026-03-18

**Soundness:** 2
**Presentation:** 2
**Significance:** 3
**Originality:** 3
**Overall Recommendation:** 3
**Confidence:** 3

**Summary:**

The paper proposes GAM, a VLA framework that tries to separate path geometry from execution speed and to separate invariant action structure from affine spatial variation. It combines an arc length based temporal alignment with a schema affine factorization and reports gains on LIBERO and SimplerEnv over several recent baselines.

**Compliance With Llm Reviewing Policy:**

Affirmed.

**Key Questions For Authors:**

Which exact theorem assumptions hold for the actual learned model, and which do not? A precise answer would affect my soundness score.
Can you provide stronger evidence that the gains come from covariance related structure rather than from better tokenization or extra supervision?
How sensitive are the results to the schema codebook and to the affine normalization choices on datasets beyond the ones shown?
Can you clarify whether the compared baselines use matched data, model size, and pretraining in every table? This would affect how I read the empirical gains.

**Limitations:**

The limitation section is too weak and the impact statement is almost empty. The paper should discuss deployment risks, failure under distribution shift, and the cost of relying on fixed symmetry assumptions.

**Strengths And Weaknesses:**

The paper has a clear high level idea and the empirical results are strong on the reported benchmarks, especially on LIBERO Long and SimplerEnv. The main strength is that the method is more structured than plain action regression and the ablations suggest both main components matter. The main weakness is soundness at the theory level. Several claims are much stronger than what is really shown, especially statements about guaranteed convergence and convexity after quotient projection. The proofs read more like intuition than full proofs and rely on simplifying assumptions that are not well tied to the actual learned system. Presentation is mixed. The motivation is clear, but the paper is very dense, overclaims often, and uses heavy terminology that sometimes makes simple ideas harder to assess. Significance is good for VLA and robot learning, since temporal and geometric invariance are relevant issues. Originality is moderate to good: the combination is interesting, but many ingredients are known, and the novelty is more in the formulation and integration than in any one component.

---

> ### Author Rebuttal · Authors · 2026-03-31
>
> ## KQ1: exact theorem assumptions
>
> Thank you for the soundness concern. We clarify that Theorem 1-2 are **conditional analyses** under simplified assumptions: **speed-reparameterization dominance** (ALP) and **nuisance spatial-variation dominance (SAF)**. Specifically:
>
> For ALP, we assume that, for the same action, a large part of inter-demonstration variation comes from execution-speed reparameterization rather than geometric-path changes. Under this assumption, Theorem 1 shows that time reparameterization acts as a nuisance factor for direct time-indexed supervision, and that **geometric-progress targets reduce the corresponding variance term**.
>
> For SAF, we assume that a substantial part of spatial variation comes from nuisance factors (reference frame, orientation, scale) rather than task-intent changes. Under this assumption, Theorem 2 shows that **quotient projection can achieve local guaranteed convergence and local convexity**, while reducing ambiguity and improving local consistency of target representations.
>
>
> ## KQ2: covariance related structure gains
>
> To address the concern that the gains may not come from covariance-related structure, we added a stricter controlled experiment. Specifically, we use the **same RVQ-VAE-style tokenization to directly model the raw action chunks** and introduce SAF beyond it. All variants are trained with matched supervision and training budget, without extra supervision.
>
> |Method|LIBERO|SimplerEnv|Real-World|
> |-|-:|-:|-:|
> |RVQVAE+SAF|95.2|54.0|62.8|
> |RVQVAE|94.7|48.6|52.0|
>
> The only difference between these two variants is what the codebook models: RVQVAE+SAF models affine-normalized canonical shapes, while RVQVAE models raw action values directly. The results show that RVQVAE+SAF improves over RVQVAE across all three settings, indicating that gains are attributable to covariance-related structure rather than better tokenization alone or extra supervision.
>
>
> ## KQ3: sensitivity of schema codebook and affine normalization choices
>
> To show that the trend is not specific to SimplerEnv, we add a cross-dataset sensitivity analysis over codebook size $K$. In each cell, values are reported as **with affine normalization / without affine normalization**:
> |K|LIBERO|Real-World|
> |-|-:|-:|
> |256|85.2/80.6|42.0/44.6|
> |512|93.7/93.0|73.5/58.1|
> |1024|96.4/95.8|76.7/65.8|
> |2048|96.2/96.0|65.3/64.9|
> |4096|95.8/94.3|70.0/56.0|
>
> These results support a clear cross-dataset conclusion: the schema codebook is neither "larger is always better" nor "smaller is always safer"; it has a stable mid-range optimum (around K=1024). Across almost all settings, the **with-affine-normalization** variant is better or comparable to the **without-affine-normalization** variant, with especially clear gains on Real-World (e.g., +10.9 at K=1024). Overall, GAM benefits from both reasonable chart granularity and affine normalization.
>
> ## KQ4: comparison fairness
>
> For fairness, all methods use **the same matched data, evaluation protocol, and the official default recommended hyperparameters of each baseline**. To further isolate gains from the model backbone and pretraining data, we additionally run a matched-setting ablation under the same Qwen2.5 backbone, training budget, and main setup, and compare Qwen2.5-GAM with Qwen2.5-FAST and Qwen2.5-BEAST.
> The detailed matched-setting results are reported in **Reviewer FUkT, W1**.
>
> Under this controlled setting, GAM remains consistently better, indicating that the improvement does not come from data mismatch or extra hyperparameter tuning.
>
> In the revision, we will substantially strengthen the Limitations and Impact sections by explicitly discussing deployment risks, failure modes under distribution shift, and the practical cost of relying on fixed symmetry assumptions.

---

> > ### Author Rebuttal · Reviewer_rLt9 · 2026-04-05
> >
> > I thank the authors for the answers!

---

> > > ### Author Response · Authors · 2026-04-07
> > >
> > > Thank you for taking the time to review our submission and read our rebuttal.
> > >
> > > If our rebuttal and clarifications lead to any further consideration during the discussion process, we would be sincerely grateful!

---

### Official Review · Reviewer_RHo7 · 2026-03-19

**Soundness:** 2
**Presentation:** 2
**Significance:** 2
**Originality:** 3
**Overall Recommendation:** 4
**Confidence:** 4

**Summary:**

This paper proposes the Generalized Action Manifold framework, highlighting the importance of enforcing temporal invariance and geometric invariance in action representations. These two types of invariance are achieved using the Arc-Length Parameterizer and the Schema-Affine-Factorization respectively, and those two methods are implemented within a VLA model. Experimental results show that the proposed methods improve performance and achieve state-of-the-art results in simulation benchmarks.

**Compliance With Llm Reviewing Policy:**

Affirmed.

**Final Justification:**

The authors' response is appreciated. Since all concerns have been addressed, the soundness rating is updated to Good and the overall recommendation to Weak Accept.

**Key Questions For Authors:**

Q1: Is it possible for the covariance matrix to exhibit multiple singular values with similar magnitudes in the datasets, leading to instability in estimating the global rotation matrix?
Q2: Does the issue described in Weaknesses 1 negatively affect the model's performance in practice?
Q3: What are the experimental results of "Robustness and Real-World Transfer"?
Q4. How does the velocity regularization encourage the generated points to maintain a valid spatial stride?
Q5. How does the model predict the arc-length?
The evaluation of the paper can be changed if the responses show that the issues of Q1 and Q2 don't exist and clarify Q3-Q5.

**Limitations:**

No. The methods could be further improved by enforcing temporal invariance in the SAF and find a better way to avoid the unstability of SVD or show that this unstability doesn't exist in the dataset.

**Strengths And Weaknesses:**

Strengths:
1. This paper clearly highlights the importance of temporal and geometric invariance in action representations.

Weaknesses:
1. The Schema-Affine Factorization does not enforce the temporal invariance, so the canonical shape remains dependent on the temporal execution speed.
2. The Schema-Affine Factorization uses SVD to calculate the global rotation matrix. However, this approach can be unstable when the covariance matrix has multiple singular values with similar magnitudes, leading to ambiguity in the resulting rotation.
3. In Line 352 the RQ3 is "Robustness and Real-World Transfer", but in Section 4.5 the RQ3 is "Ablation of Theoretical Components", and the paper does not show the experimental result of "Robustness and Real-World Transfer" in other places.
4. It is unclear why the velocity regularization can prevent the model from converging to trivial stationary solutions and encourages the generated points to maintain a valid spatial stride.
5. In Section 3.3.1 the arc-length is calculated using the predicted action chunk, but in Section A.2 it seems like the arc-length is predicted using the Dynamics Head, which is not consistent with Section 3.3.1.

---

> ### Author Rebuttal · Authors · 2026-03-31
>
> ## KQ1&W2:
> To address the concern that canonicalization may become unstable, we add an SVD-based canonicalization stability analysis. We compute singular-value gaps of trajectory covariance and report the near-degenerate ratio:
> $$ \mathrm{Near Degenerate Ratio}(\tau)=\frac{1}{N}\sum_{i=1}^{N} \mathbf{1}\!\left(\Delta_{12}^{(i)}<\tau \;\text{or}\; \Delta_{23}^{(i)}<\tau\right), $$
> where $\Delta_{ij}=\frac{\sigma_i-\sigma_j}{\sigma_i}$ and $\sigma_i$ is the $i$-th singular value of the covariance matrix.
>
> Results show that most samples have a clear dominant motion axis; ambiguity is mostly in weaker secondary axes, and near-degenerate cases are uncommon.
>
> |Dataset|Mean $\Delta_{12}$↑|Mean $\Delta_{23}$↑|Near-Degenerate Ratio at $\tau=0.02$↓|
> |-|-:|-:|-:|
> |LIBERO|0.92|0.07|0.03|
> |SimplerEnv|0.79|0.15|0.08|
> |Real-World|0.88|0.13|0.04|
>
>
> ## KQ2&W1:
> In GAM, SAF extracts the **geometric shape of an atomic action** after factoring out spatial execution parameters (shown in line 232-240). SAF is not the module that enforces temporal invariance; that role is handled by ALP at the action-chunk level. Absorbing timing variation into SAF would weaken semantic action decisions and blur where motions should be fast or slow. Therefore, this design choice does **not** materially harm model performance in practice.
>
> Empirically, this design is beneficial: in an alternative SAF design where the codebook is built by **truncating trajectories using the average arc length of the same atomic action**, performance drops notably (LIBERO: 96.4%→92.5%, SimplerEnv: 56.3%→48.0%). This indicates that the current SAF decomposition better captures task-level geometry and execution pace, rather than imposing temporal invariance at the action schema stage.
>
> Notably, in LIBERO and SimplerEnv, an atomic trajectory coincides with the full demonstration trajectory. For longer-horizon tasks, trajectories would be segmented into atomic sub-trajectories.
>
>
> ## KQ3&W3:
> We apologize for missing results and add real-robot experiments on Agibot G1 with three tabletop tasks of increasing difficulty:
>
> - ClickKeyboard: keyboard clicking;
> - ShakeCup: shaking a cup;
> - ScoopRice: scooping rice.
>
> Under the same backbone and matched setting, we compare Qwen2.5-GAM, Qwen2.5-FAST, and Qwen2.5-BEAST (see the real-robot result table in **Reviewer FUkT, W2**).
>
> As shown in the table, Qwen2.5-GAM consistently outperforms the baselines across all three real-robot tasks, and mean±std over repeated runs indicates stable gains rather than task- or seed-specific luck. This supports robust transfer of temporal+geometric disentanglement from simulation to real robots.
>
>
> ## KQ4&W4:
> We thank the reviewer for pointing this out and apologize for the confusion. We adopt the following precise form:
>
> $$
> L_{\mathrm{vel}}=\frac{1}{T-1}\sum_{i=1}^{T-1}\big(\mathrm{L2norm}(\xi_i^{\mathrm{pred}}-\xi_{i-1}^{\mathrm{pred}})-\mathrm{L2norm}(\xi_i^{\mathrm{gt}}-\xi_{i-1}^{\mathrm{gt}})\big)^2
> $$
>
> This term directly **constrains the displacement magnitude between consecutive predicted points to match the ground-truth valid spatial stride**.
>
> The motivation is this limitation: ALP replaces absolute-time alignment with geometric-progress alignment, so training focuses on path geometry rather than fixed speed. With only path-level supervision, predictions can become **over-smoothed and average out fine-grained fast/slow variations**. $L_{vel}$ compensates for this by matching local stride patterns of real demonstrations, especially for real-robot data with semantically meaningful pauses or slow-motion phases.
>
> This is particularly important for real-robot data, where semantically meaningful pauses or slow-motion phases are common. Without $L_{vel}$, these details can be over-smoothed by simple interpolation along the canonical shape.
>
>
> ## KQ5&W5:
> We first clarify that **GAM does not contain any explicit arc-length prediction module**.
>
> In Sec. 3.3.1, the arc-length in Eq. 4 is not predicted by any network. It is computed from the current predicted action chunk and used only to construct geometric-progress supervision for the flow-matching target in Eq. 10.
> By contrast, Appendix A.2 introduces affine parameters as auxiliary supervision on top of action-intent prediction. There, the Dynamics Head predicts $z_{\text{time}}$, a scalar temporal factor associated with the action schema, representing **trajectory duration**.
>
> Overall, ALP is used only to generate equal-arc-length supervision for Executor.

---

> > ### Author Rebuttal · Reviewer_RHo7 · 2026-04-03
> >
> > KQ1-3 and KQ5 have been fully resolved. However, regarding KQ4, it seems that the gradient of L_vel needs to be backpropagated through the denoising process during training. Does this lead to high computational cost or GPU memory usage?

---

> > > ### Author Response · Authors · 2026-04-03
> > >
> > > Thank you for the follow-up question. This is indeed an important concern: since `L_vel` is defined on the local stride of the predicted trajectory, its gradient does need to be backpropagated through the denoising process during training. To directly quantify its impact on training efficiency, we further provide a training-overhead analysis and report the following metrics:
> > >
> > > - training latency per step (ms)
> > > - throughput (samples/s)
> > > - peak GPU memory usage (GB)
> > > - relative overhead compared to training without `L_vel`
> > >
> > > The results are summarized below:
> > >
> > > | Setting                          | Training Latency (ms) / step | Throughput (samples / s) | Peak GPU Memory (GB) |
> > > | -------------------------------- | :--------------------------: | -----------------------: | -------------------: |
> > > | w/o `L_vel` |            948.1             |                    152.0 |                50.08 |
> > > | w/ `L_vel`  |            961.7             |                    149.8 |                50.92 |
> > >
> > > From these numbers, adding `L_vel` increases the per-step training latency by only 13.6 ms (from 948.1 ms to 961.7 ms), which corresponds to a relative overhead of about 1.4\%. The throughput drops only slightly, from 152.0 to 149.8 samples/s, i.e., by 2.2 samples/s or about 1.4\%. The peak GPU memory increases from 50.08 GB to 50.92 GB, which is an absolute increase of 0.84 GB (about 1.7%). Overall, all measured overheads remain below 2%, indicating that the additional cost of `L_vel` is limited rather than burdensome. More importantly, **this limited training overhead brings a meaningful modeling benefit**. `L_vel` explicitly constrains the local stride / velocity profile, which helps prevent over-smoothed trajectories and trivial local stagnation.
> > >
> > > Taken together, we believe that `L_vel` achieves a reasonable trade-off between training efficiency and modeling benefit: it introduces only a very small additional training cost, while improving the faithfulness of local execution dynamics, and it incurs no extra inference-time overhead.
> > >
> > > If this additional analysis addresses your concern, we would greatly appreciate your reconsideration of the paper’s soundness and overall rating. We also sincerely thank you for your constructive feedback and effort you devoted to carefully evaluating our work!

---

### Official Review · Reviewer_FUkT · 2026-03-24

**Soundness:** 3
**Presentation:** 4
**Significance:** 3
**Originality:** 3
**Overall Recommendation:** 5
**Confidence:** 4

**Summary:**

This paper proposes a novel framework called General Covariant Action Modeling (GAM), aiming to address the prevalent issues of 'mode averaging' and 'temporal misalignment' in current robot imitation learning. Through structured disentanglement, the GAM framework introduces 'temporal invariance' and 'geometric invariance,' successfully separating the topological structure of actions from their execution parameters. The experimental results also validate the authors' proposed viewpoints.

**Compliance With Llm Reviewing Policy:**

Affirmed.

**Final Justification:**

Thanks for the authors rebuttal! The updated result solved my concern and i will raise my score to 5.

**Key Questions For Authors:**

Please refer to the weaknesses.

**Limitations:**

Please refer to the weaknesses.

**Strengths And Weaknesses:**

Strength:

1.  The authors delve deeply into the issues of mode averaging and temporal misalignment in imitation learning, proposing theoretical solutions grounded in concepts like general covariance and Lie groups. Theorems 1 (Diffeomorphism Invariance) and 2 (Convexity in Quotient Space) provide a solid theoretical foundation.
2.  The GAM framework achieves temporal invariance through the "Arc-Length Parameterizer" (ALP), decoupling spatial geometry from temporal dynamics. It achieves geometric invariance via "Schema-Affine Factorization" (SAF), mapping trajectories to canonical "world lines." This dual invariance design is highly innovative, and results on benchmarks like LIBERO and SimplerEnv demonstrate its effectiveness.

Weakness:

1.  There appear to be some issues with the baseline results presented by the authors. Models based on Qwen2.5-VL-3B-Instruct, particularly with various improvements like those seen in StarVLA, could potentially achieve results on LIBERO that surpass the authors' current enhanced solutions. Could the authors provide further explanation for this discrepancy, or investigate whether further post-training on a stronger base model would yield improvements?
2.  A lack of real-world experimental setup is noted. It is unclear whether this approach demonstrates consistent effectiveness on physical robots, as the modes of actions collected from real-world data acquisition can differ from those planned in simulation.

---

> ### Author Rebuttal · Authors · 2026-03-31
>
> ## W1:
> To address concerns about discrepancies in StarVLA results, we reproduced the experiments using the official StarVLA configuration, and conducted a matched-setting comparison among Qwen2.5-GAM, Qwen2.5-FAST, and Qwen2.5-BEAST under the same backbone, training budget, and main hyperparameter setup. The results are shown below.
>
> |Method|Spatial↑|Object↑|Goal↑|Long↑|Avg.↑|
> |-|-:|-:|-:|-:|-:|
> |Qwen2.5-GAM|98.6±0.2|97.4±0.1|96.7±0.1|91.9±0.3|96.2±0.2|
> |Qwen2.5-FAST|97.1±0.3|96.9±0.1|94.8±0.2|89.2±0.1|94.5±0.2|
> |Qwen2.5-BEAST|94.3±0.3|96.1±0.5|93.9±0.1|88.4±1.1|93.2±0.5|
>
> |Method|PutSpoon↑|PutCarrot↑|StackBlock↑|PutEggplant↑|Avg.↑|
> |-|-:|-:|-:|-:|-:|
> |Qwen2.5-GAM|65.3±2.4|43.1±4.2|29.2±0.0|93.1±2.4|57.7±2.4|
> |Qwen2.5-FAST|45.8±4.2|37.5±4.2|18.5±7.9|79.2±7.2|45.3±5.9|
> |Qwen2.5-BEAST|40.3±2.4|33.3±8.4|16.7±4.2|77.8±8.7|42.2±5.9|
>
> On LIBERO, Qwen2.5-GAM achieves an average success rate of 96.2±0.2, outperforming Qwen2.5-FAST (94.5±0.2) and Qwen2.5-BEAST (93.2±0.5). On SimplerEnv, Qwen2.5-GAM reaches 57.7±2.4, substantially higher than Qwen2.5-FAST (45.3±5.9) and Qwen2.5-BEAST (42.2±5.9). These results suggest that, **under a unified Qwen2.5 backbone and matched recipe, the gains of GAM do not come from using a stronger backbone or a larger training budget, but from its structured temporal + geometric action representation**.
>
> At the same time, we acknowledge that our reproduced numbers still show some discrepancy from the reference results reported in the official repository. This phenomenon is largely consistent with the reproduction gap reported by two recent StarVLA-based works recognized by the official repository: LangForce [1] and TwinBrainVLA [2]. Therefore, we believe that this discrepancy more likely reflects the implementation/reproduction variance of the StarVLA pipeline across environments, rather than an unfair comparison introduced by our method.
>
> To further ensure result stability, we repeated all experiments with seeds 1, 2, and 3, and reported mean ± std in the table. As can be seen, Qwen2.5-GAM consistently achieves higher average performance on both LIBERO and SimplerEnv while maintaining acceptable variance. This indicates that our conclusions are based on stable gains reproducible across multiple random seeds.
>
> [1] Lian S, Yu B, Lin X, et al. LangForce: Bayesian Decomposition of Vision Language Action Models via Latent Action Queries. arXiv e-prints, 2026: arXiv:2601.15197.
>
> [2] Yu B, Lian S, Lin X, et al. TwinBrainVLA: Unleashing the Potential of Generalist VLMs for Embodied Tasks via Asymmetric Mixture-of-Transformers. arXiv preprint arXiv:2601.14133, 2026.
>
>
>
> ## W2:
> We apologize for the missing results and have added real-robot experiments in the revised version. Specifically, we evaluate three tabletop manipulation tasks of increasing difficulty on the Agibot G1 robot platform. These tasks are intentionally **contact-rich and long-horizon**, where action-mode distributions differ from simulation rollouts:
> - ClickKeyboard: keyboard clicking, testing short-horizon localization and contact stability;
> - ShakeCup: shaking a cup, testing continuous trajectory control and pose stability;
> - ScoopRice: scooping rice, testing contact-rich fine manipulation and long-horizon execution stability.
>
> Under the same backbone and matched setting, we compare Qwen2.5-GAM, Qwen2.5-FAST, and Qwen2.5-BEAST under a unified real-robot protocol (same sensing setup, control frequency, horizon, and success criterion), and report mean±std over repeated runs. The results are shown below.
>
> |Method|ClickKeyboard↑|ShakeCup↑|ScoopRice↑|Avg.↑|
> |-|-:|-:|-:|-:|
> |Qwen2.5-GAM|76.7±4.7|86.7±9.4|66.7±4.7|76.7±6.3|
> |Qwen2.5-FAST|66.7±9.4|80.0±8.2|53.3±12.5|66.7±10.0|
> |Qwen2.5-BEAST|53.3±4.7|50.0±8.2|26.7±11.5|43.3±8.1|
>
> As shown in the table, Qwen2.5-GAM consistently outperforms the baselines on all three real-robot tasks. GAM improves over FAST by +10.0 (ClickKeyboard), +6.7 (ShakeCup), and +13.4 (ScoopRice), and over BEAST by +23.4, +36.7, and +40.0, respectively. The gains remain positive across tasks with different difficulty levels, including the hardest contact-rich task (ScoopRice), which demonstrates the robustness of GAM. The mean±std results from repeated runs further show stable improvements rather than task-specific or seed-specific luck. This indicates that **the proposed temporal + geometric disentanglement not only works in simulation, but also transfers robustly to real-robot manipulation**.

---

> > ### Author Rebuttal · Reviewer_FUkT · 2026-04-04
> >
> > Thank you for your thoughtful response. Your conclusions on starvla are indeed quite convincing.
> >
> > However, regarding the LIBERO benchmark, I note that numerous recent works have achieved performance levels approaching 99.0+. While I acknowledge that such comparisons have certain limitations, some of the experimental results on LIBERO do give me pause. This raises my primary concern: whether the work can demonstrate substantial improvements in practical effectiveness beneath its elegant theoretical framework.
> >
> > On the other hand, I am pleased to see the real robot experimental results provided by the authors. For robot research, real-world validation is an indispensable component. I sincerely hope the authors will fully present this content in the final version.
> >
> > Overall, I still tend to hold a positive view and keep my sorce.

---

> > > ### Author Response · Authors · 2026-04-06
> > >
> > > **Thank you again for the constructive feedback and for maintaining a positive overall view of our paper.**
> > >
> > > We believe the discrepancy between our results and many reported VLA results mainly comes from two factors: **(i) our setting does not introduce extra sources of information beyond the Qwen2.5-based setup, and (ii) there is often no unified matched setting across methods for a fair comparison.**
> > > To address **(i)**, we adapt GAM to some of the most widely used VLA backbones and demonstrate the portability of GAM to existing VLA backbones. To address **(ii)**, we show that under the StarVLA framework, different hyperparameter settings can also produce highly competitive results. Finally, to further demonstrate the reliability of GAM, we provide additional results on CALVIN.
> > >
> > > (1) **Our goal is not to introduce a new standalone VLA model, but to provide a better supervision criterion that can plug into existing VLA families.** To make this point explicit, we additionally incorporate GAM into representative pretrained VLA backbones. For fairness, we use the corresponding recommended configurations for each backbone, i.e., pi0.5: 30k steps with 256 batch size, and GR00T-N1.6: 20K steps with batch size 640.
> > >
> > > | Model | Spatial | Object | Goal | Long | Average |
> > > | --- | --- | --- | --- | --- | --- |
> > > | pi0.5 | 98.80 | 98.20 | 98.00 | 92.04 | 96.85 |
> > > | **pi0.5 + GAM** | 99.55 | 98.80 | 98.25 | 93.10 | 97.43 |
> > > | GR00T-N1.6 | 97.65 | 98.45 | 97.50 | 94.35 | 96.99 |
> > > | **GR00T-N1.6 + GAM** | 97.70 | 99.30 | 97.80 | 95.20 | 97.50 |
> > >
> > > These added experiments are intended to clarify the scope of our contribution. **GAM is meant to be a portable model-agnostic supervision design that improves action modeling**, rather than a gain tied to one custom architecture. This is also why we believe the fairest evaluation is the matched-setting comparison in our main paper: it isolates the effect of the supervision target itself.
> > >
> > > (2) **Higher LIBERO scores can indeed be obtained under overfitting settings.** All LIBERO experiments in our paper were intentionally conducted under the default StarVLA configuration. In particular, our standard matched-setting result is obtained with **30k training steps**, because our goal was to isolate the contribution of the learning target rather than maximize the absolute benchmark number through additional benchmark-specific tuning. For reference, the following results illustrate how LIBERO scores can be further increased when the training is extended to an overfitting-oriented setting with **100k steps**:
> > >
> > > | Model | Spatial | Object | Goal | Long | Average |
> > > | --- | --- | --- | --- | --- | --- |
> > > | Qwen2.5-GAM | 98.6 | 97.4 | 96.7 | 91.9 | 96.2 |
> > > | Qwen2.5-GAM overfitting | 99.1 | 99.4 | 98.3 | 97.4 | 98.6 |
> > >
> > > This comparison makes the benchmark saturation effect more explicit. Under the default matched StarVLA setting, Qwen2.5-GAM reaches an average success rate of 96.2. When we deliberately continue training toward an overfitting regime, the average score further rises to **98.6**. **This shows that LIBERO scores can be pushed substantially higher through stronger benchmark-specific fitting, even without changing the core method.** Therefore, while such near-ceiling numbers are achievable, they should not be treated as a clean proxy for method quality unless the backbone and training budget are strictly matched.
> > >
> > > (3) **Our method remains consistently effective beyond LIBERO, as verified on CALVIN.** Under the same matched setting, we also observe stable gains on CALVIN, which is a substantially different long-horizon benchmark. The results are summarized below:
> > >
> > > | Method | ABC->D | ABCD->D | Avg. |
> > > | --- | --- | --- | --- |
> > > | Qwen2.5-GAM | **4.35 +/- 0.08** | **4.51 +/- 0.03** | **4.43 +/- 0.06** |
> > > | Qwen2.5-FAST | 4.10 +/- 0.18 | 4.25 +/- 0.05 | 4.18 +/- 0.12 |
> > > | Qwen2.5-BEAST | 3.88 +/- 0.02 | 4.24 +/- 0.16 | 4.06 +/- 0.09 |
> > >
> > > These results show that the gain is not confined to a single benchmark. **Qwen2.5-GAM achieves the best result on both CALVIN splits and improves the average score by +0.25 over FAST and +0.37 over BEAST**, which supports our claim that the proposed supervision target leads to more stable long-horizon learning rather than merely exploiting a particular LIBERO evaluation protocol.
> > >
> > > **Overall, we would like to emphasize again that GAM is a more transferable and more effective supervision target, not merely a benchmark-specific optimization trick.**

---

### Decision · Program_Chairs · 2026-04-30

**Decision:**

Accept (regular)

**Comment:**

Multiple reviewers found the core idea well motivated and the framework technically solid. After rebuttal two reviewers recommend clear acceptance and a third recommends weak accept, while the fourth votes for a weak-reject, resting on theoretical concerns. Overall, the AC agrees with the general positive consensus on the paper and notes that the reviewer recommending weak rejection does not provide substantiative comments with their rebuttal acknowledgment, stating that concerns have been partially addressed. Considering the general positive consensus, and that claims on limited theoretical analysis are in contrast with observations by other reviewers, the AC recommends acceptance.